# Big Data in Smart City: Management Challenges

**Mladen Amović** [1],*, **Miro Govedarica** [2] , **Aleksandra Radulović** [2] and **Ivana Janković** [1]

[1] Faculty of Architecture, Civil Engineering and Geodesy, University of Banja Luka, 78000 Banja Luka, Bosnia and Herzegovina; ivana.jankovic@aggf.unibl.org

[2] Faculty of Technical Sciences, University of Novi Sad, 21000 Novi Sad, Serbia; miro@uns.ac.rs (M.G.); sanjica@uns.ac.rs (A.R.)

* Correspondence: mladen.amovic@aggf.unibl.org

**Abstract:** Smart cities use digital technologies such as cloud computing, Internet of Things, or open data in order to overcome limitations of traditional representation and exchange of geospatial data. This concept ensures a significant increase in the use of data to establish new services that contribute to better sustainable development and monitoring of all phenomena that occur in urban areas. The use of the modern geoinformation technologies, such as sensors for collecting different geospatial and related data, requires adequate storage options for further data analysis. In this paper, we suggest the biG dAta sMart cIty maNagEment SyStem (GAMINESS) that is based on the Apache Spark big data framework. The model of the GAMINESS management system is based on the principles of the big data modeling, which differs greatly from standard databases. This approach provides the ability to store and manage huge amounts of structured, semi-structured, and unstructured data in real time. System performance is increasing to a higher level by using the process parallelization explained through the five V principles of the big data paradigm. The existing solutions based on the five V principles are focused only on the data visualization, not the data themselves. Such solutions are often limited by different storage mechanisms and by the ability to perform complex analyses on large amounts of data with expected performance. The GAMINESS management system overcomes these disadvantages by conversion of smart city data to a big data structure without limitations related to data formats or use standards. The suggested model contains two components: a geospatial component and a sensor component that are based on the CityGML and the SensorThings standards. The developed model has the ability to exchange data regardless of the used standard or the data format into proposed Apache Spark data framework schema. The verification of the proposed model is done within the case study for the part of the city of Novi Sad.

**Keywords:** smart city; geospatial big data; Apache Spark SQL; sensors

## 1. Introduction

In recent years, information technologies are more dynamic since applications and services moved their limits. There is a huge need to store and process structured, semi-structured, and non-structured data and offer results to different applications according to their requirements. The significant extent for this improvement is the need to make decisions based on accumulated knowledge through the new way of collecting and processing information necessary for decision-making procedures. Digitalization of all processes, spatial referencing of the phenomena that surround us, as well as increased urbanization lead to the need for optimization of the processes in urban space. According to the United Nations research, in 2018, 55.3% of the human population lived in urban areas, which is estimated to become 60.4% in 2030 [1,2].

These facts as well as the heterogeneity of the Earth's population indicate that there is a need for new approaches and strategies in planning and providing a quality life through the concepts of agricultural production planning, health care, traffic optimization, etc. The development of the concept of smart cities provides an opportunity for the continued usage

of obtained data with the ultimate goal of solving problems. Smart cities involve a large number of data inputs (sources, sensors) which have different standardization principles. The collection of different types of data from heterogenous sources gives a better profile for final decision-making procedures, defined by the basic mathematical principles of multicriteria analysis. Geoinformation and communication technology (GeoICT) is being increasingly adopted to foster urban sustainability and smart cities [3].

Smart cities are an important concept of life development for the whole population. Smart cities are working under well-defined standardized models, mainly defined by ISO series of standards. Salina et al. [4] and Moniruzzaman et al. [5] identified that there are many stores dealing with the "smart data", but mainly they aim to solve some specific problems and provide returns. Sivarajah et al. [6] identified visualization as one of the big data problems. It is an important segment of the smart city in the process of the publication of results and the ways of communicating with a user. There is an intention to every phenomenon measured in the smart city environment that can be visualized with geospatial referencing. By the recommendations of the United Nations smart city strategy for sustainable development, point cloud is identified as a geospatial representation tool. In the context of smart cities, there are many different sensor sources which provide data for smart cities and decision-making processes. Those sources are rarely from the same stores as geospatial data and usually differ in use standards. On the implementation level, smart cities have a problem in combining those data sources in the context of avilable data models and tools. CityGML as a standard provides effective implementation of the identified system, which includes geospatial and sensor data. The CityGML model has great potential for extensions and adaptations for specific problems [7,8]. CityGML integrates geospatial representation in the LoD0–LoD4 levels of details together with sensor measuraments. Indoor space is represented on the LoD4 level but has limitations for some important analyses, such as a problem of navigation through indoor space [9]. Kim et al. [10] and Ryoo et al. [9] suggest IndoorGML as an adequate solution for these.

The goal of this paper is the possibility to present all "smart data" in one integral model which connects them and forms a smart city store with the potential for complex data manipulations. Smart data collected from different sensors are often heterogeneous. There are several standards in the context of working, collecting, and storing data [11]. Most of them are based on the definitions of the Internet of Things standard and the SensorML standard. In the last decades, the size of data increased tremendously to the range of petabytes. Database management systems are challenged to handle such huge data volumes, especially geospatial data as a complex data type. Availability of a large number of data sources indicates importance of big data in supporting the smart city applications and services. Al Nuaimi et al. [12] investigated opportunities for utilizing big data in the smart city. They defined that main problems of big data in the context of smart cities can be shown through the five V definitions: value, velocity, variety, variability, and volume:

- data are generated in real time, which leads to the tremendous amount of data to be dealt with—volume (identified by [13,14]);
- data are collected from different sources (smart phones, computers, GNSS, enviromental sensors, cameras)—variety [6];
- big data are catalogued and stored in different platforms that often cause a problem in that some data are unused—velocity [15];
- smart city demands change in the approach of performing the complex analytics operations over such huge amount of data—velocity [16];
- big data systems must provide the data that are included in analytical processes with a focus on enhancing smart city applications—variability [14];
- a need to transform a huge amount of data into business based on big data collection, management, and analysis—value [12,17,18].

In the context of defined problems, limitations of the smart city based on the relational database systems are:

- volume—related to the storage performances and the performances of the platform where the system is installed,
- variety—the conceptual model of the smart city solution does not have a possibility to add different sources to the platform because they work on different standard protocols,
- velocity—most of sources have their own storage,
- varibility—a problem to perform complex analysis with data used from different sources,
- value—same as volume, limited with the storage and the platform performances.

Geospatial data are analyzed in the context of big data in terms of collecting, storing, and processing large amounts of geospatial data. The management context of decision-making processes places big data as an additional building block in the system of exchange and processing of large amounts of geospatial data more efficiently than traditional solutions [19]. The amount of collected data largely exceeds the possibility of their storage on individual computers and requires storage on a cluster of computers [20]. Traditional relational database management systems have problems to handle all these data together, especially if big data concepts are considered. The 3D city models are an integral part of the smart city [21].

In this paper, GAMINESS is described as a new management system based on big data concepts, which provide management over different structured, semistructured, and non-structured smart city data. The system consists of two components: a geospatial component and a geosensor component. GAMINESS provides connectivity of different data sources regardless of the standardization rules and the data formats. The main advantage of such a system is that it represents a solution for the variety problems in big data. GAMINESS also speeds up performances vertically, allowing fast responses for the data requested by clients.

The goals of this paper were:

- migrating/transferring the standard smart city concept defined as the relational DB system to the big data framework;
- creating the GAMINESS conceptual model based on the CityGML 3.1.1 standard extended with the IndoorGML standard, the IoT standard, and the DigtialTwins smart city recommendations and extended with the connection protocols for different data sources and complex data types based on data sources;
- developing the model which integrates all data in one structure based on a map-reducing paradigm of big data.

The paper is structured as follows: after the introduction in Section 1 and the related work review in Section 2, Section 3 presents a methodology in the form of a roadmap that was used to develop the model of the GAMINESS management system. Section 4 presents the implementation of GAMINESS into the big data framework. In Section 5, the case study for the city of Novi Sad is presented. Conclusions and future work are discussed afterward.

## 2. Related Works

Smart cities cover different phenomena of life and must include all of them to provide expected feedback, such as: Internet of Things, internet, development of smartphones, computers, and other mobile communication networks necessary for smart cities, defined big data problems, and created problems of speed, structure, volume, cost, value, security, privacy, and interoperability. Traditional methods are impotent when facing big data problems because of their lack of scalability and efficiency. All those problems in the context of smart cities are defined as the five V problem.

(a)     Volume problems and solutions

Big data analysis requires large dataset analysis and the creation of advanced analytic algorithms that require a great deal of processing power and resources. By adopting virtualization, software frameworks become more efficient. Existing spatial database systems (DBMSs) extend relational DBMSs with new data types, operators, and index structures to handle spatial operations based on the open geospatial consortium standards [22]. Even

though such systems provide full support for spatial data in some ways, they suffer from a scalability issue. This happens because the massive scale of available spatial data hinders the process when using traditional spatial query processing techniques. Recent papers such as Zhou et al. [23] and Boehm et al. [24] extended the Hadoop ecosystem to perform spatial analytics at scale. The Apache Spark module was used to complete the parallelization for the complex data types. Apache Spark adopts a flexible resilient distributed dataset (RDD) programming model which is originally positioned as a fast and general data processing system. To work with these data, there are several big data oriented storage options such as HDFS, HBase, Kudu, Kafka, etc. Additionally, there are some research papers where Apache Spark as a big data engine was used to connect to DBMS and process spatial data stored in those DBMSs [25]. There are many big data based solutions dealing with processing large amounts of data. The main problems in the distributed processing of geospatial big data are the methods for indexing and storing data, which are the focus of this paper. With the use of the GeoSpark library, the standard form of RDD was expanded into a spatial RDD (SRDD) form (20). The query operations, provided through a spatial query processing layer, were performed over objects stored in SRDD [25]. GeoMesa provided the spatio-temporal indexing for massive storage of point, line, and polygon data as well as near real-time stream processing of spatio-temporal data by layering spatial semantics on top of Apache Kafka. Currently, it works with three major Hadoop-based database systems: Accumulo, Apache HBase, and Google Cloud Bigtable [26]. The GeoWave library supports Apache Accumulo and Apache Hbase repositories and provides out-of-the-box support for distributed key-value stores. It uses multiple gridded space filling curves (SFCs) to index data to the desired key-value store [27]. Indexing information is stored in a generic keystructure that can be used for server-side processing [28]. Zhou et al. [23] proposed a GeoSpark SQL framework which provides convenient interface for geospatial SQL queries on top of Apache Spark. The experimental results showed that Apache Spark had a better performance than traditional DBMSs for various types of geospatial queries [26]. The methods based on Apache Spark for large point cloud management are described in [29,30]. The method for ingesting the point clouds in the Apache Spark data structures is presented in [29]. The indexing of point clouds based on space filling curves is presented in [30]. The methods for classification, feature identification, and change detection using large point clouds are described in [26,27,29,31,32]. A file-centric approach for storage of large point clouds collected by LiDAR is described by Boehm et al. [33]. Rectangular regions are indexed using the Geohash system and stored in MongoDB database along with the location of a corresponding file. Such a structure allows executions of MapReduce operations on point clouds, either from MongoDB or from an external framework such as Apache Hadoop [27]. Pajic et al. [27] described a model of a point cloud data mangement system (PCDMS) fully based on the big data paradigm which would allow practically unlimited scalability of the system.

(b)     Variety problems and solutions

Considering geospatial representation of the smart city, there is a great deal of research that provided solutions for problems of variety in the context of big data. Hyung-Gyu et al. [9] compared CityGML Lod4 and IndorGML models. As a result, CityGML was defined as a feature model, with supported visualization, excellent performances for geometry analysis, poor performances for route analysis, and medium performances for context analysis. IndoorGML was defined as a cellular space model, with low supported visualization, poor performances of geometric analysis, and excellent performances of route and context analysis. According to this, both standards have some advances. Kim et al. [10] proposed methods for automatic derivation of IndoorGML data from a CityGML LoD4 dataset and external references from an IndoorGML instance to an object in CityGML data [10]. Claridades et al. [34] proposed an approach to integrate IndoorGML with Indoor POI data by extending the IndoorGML notion of space and topological relationships.

In smart cities, data are generated from multiple sources and have multiple formats, i.e., structured, unstructured, and semi-structured. Storing and processing such data is not

possible using traditional software, thus data formats and sources need to be considered and factored into the solution while designing the smart city applications where big data analytics have their chance [16]. The Internet of Things as a technique for the smart city collects a large amount of data using a large number of sensors [35]. The inclusion of sensor technologies is well covered by the OGC Sensor Web Enablement initiative (SWE) [36]. SWE is a set of standards which not only allows modeling of sensor descriptions and observations but also specifies web services to exchange sensor descriptions and observations in an interoperable way. The combination of open standards (and APIs) eases the delivery of geospatial features and sensor observations in a coherent way and thereby supports interoperable and cross-domain city services [37]. The CityGML model is not able to represent time-dependent and dynamic properties. It allows storing properties as static values. This lack was described as a technological gap by Huang et al. [38]. In the same paper, the authors proposed open and interoperable SW-IoT end-to-end architecture based on the OGC SensorThings API. Still, there is a limitation of representation of IndoorGML in the context of big data. Gaur et al. [39] used Apache Spark big data as a processing tool over smarthphone accelerometere datasets. Saraswathi et al. [40] used Apache Spark as big data engine for real time traffic monitoring system to predict the total traffic count of streaming data in various routes to reduce traffic congestion. IoT applications face unique problems related to their widely distributed, resource-constrained device endpoints [41].

(c)     Velocity problems and solutions

Today, there are several DBMS systems which are used for data storage, which represents smart cities. The 3DCityDB is one of the most used solutions. It works with Oracle and PostgreSQL DBMS. The 3DCityDB Importer/Exporter has a built-in mechanism that was developed for importing and exporting city data in different formats. The 3DCityDB is based on CityGML standard, but one of the main problems is a lack of the possibility to represent indoor space using the IndoorGML standard, which is used for navigation. Data stored in other models such as IFC and CityGML can be converted into IndoorGML. Biljecki et al. [42] noted that most available 3D city models contain many geometric and topological errors. The most common errors are related to incomplete surfaces, duplication of vertices, self-intersecting volumes, etc. Additinally, there is the problem with storage of sensor data collected from different sensors. Several papers cover integration of city data, mainly stored as CityGML [43,44]. Such solutions are often only visualization tools, and data are stored separately [22]. Other solutions are provided by the 52° North project as a JavaScript library for visualizing the semantical city model stored in CityGML and the OGC SOS observation service from 52° North [45]. CityJSON is a JSON-based encoding system for storing 3D city models, also called digital maquettes or digital twins [46].

(d)     Variability problems and solutions

Point cloud data can be used for different types of analysis. Apache Spark introduces a new platform for processing on a cluster of computers that runs on one of the cluster's operating systems. Apache Spark is designed to maintain scalability and tolerance for MapReduce algorithms, which is the main concept of big data DBMS. Some research in the field of processing large amounts of point clouds using cloud computing technology was given by Liu et al. [31].

(e)     Value problems and solutions

Big data technologies are used to optimize storage, processing, post queries, and analysis for all types of intelligent applications and services. 52° North is a platform which offers a list of services through adequate SOSs and provides geostatistic analysis. Another platform that enables connection of big data and geostatistical analysis to the smart city platform is DABAMOS. It also enables inclusion of different types of geodetic equipment, which provides permanent data. The OpenADMS node stores observation data in the PostgreSQL relation database [47]. Moreover, there is an increasing need to conduct real-time storage, processing, query, and analysis for big data. There are many smart city intelligent city map applications such as Smart City Map, City Drop, Jobs-

Nearby, and Hexagon applications for fusing automation and IoT with real-time analytics capabilities. In the context of the data storage for smart city solutions due to the diversity and the dynamism of its sources, where the time series in DB are frequently used, and InfluxDB [48] and Druid [49] are used as well. Smart city solutions need to have adequate, real time notifications on events, thus the platform must have a stream analytics engine which reacts upon the event and sends notifications. Node-RED is one such tool for wiring together hardware devices, APIs, and online services. PubNub is a data stream network that offers a service for the user to design their own architecture [50,51]. AWS IoT is a platform that enables connection of devices and interaction with them even when they are offline [52].

## 3. Materials and Methods

Big data technologies are not a unique technology, but a combination of new and old technologies that help enable data management of large amounts of different data in structured, semi-structured, and unstructured forms. Figure 1 shows possiblities of the proposed GAMINESS management system. It integrates loading and storing data from different sources regardless of their structure according to the five V rules. Sources are connected to the management system as separate storages. This provides the velocity objective. Traditional systems only receive data which are structured according to a predefined concept such as CityGML. GAMINESS can recive data from sources without variablity problems. It uses algorithms to read such structures and map data by using user defined types. In traditional systems, the speed of transactions directly affects the performance of the system. The GAMINESS management system is characterized by big data processing using a cluster network, which speeds up performance vertically and improves overall system performance.

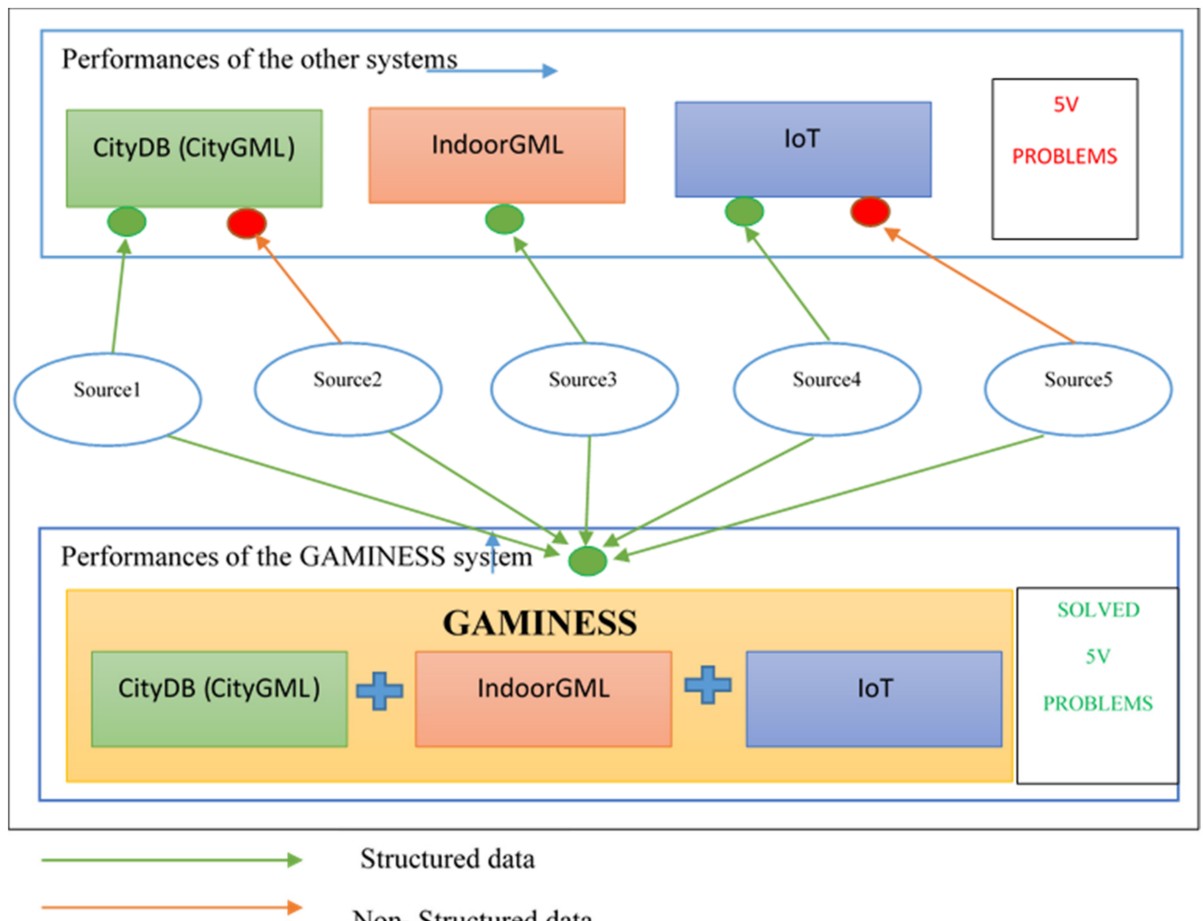

**Figure 1.** Migration from standard city model to the GAMINESS big data model.

The paper presents a proposal for a new solution that provides a different approach of support for the mentioned five characteristics compared to the most commonly used management systems such as PostgreSQL and Oracle. The research methodology in the form of a roadmap is shown in Figure 2. The proposed model, based on the concepts of the big data paradigm and using process parallelization algorithms for distributed sources, gives significant efficiency over large amounts of data stored (petabytes and larger). The advantage of the big data system is primarily reflected in the processing of large amounts of data, while, with smaller amounts of data, this advantage is not noticeable, and thus there is no particular need to use it. By sharing data through network system clusters, these a6lgorithms provide higher processing speeds, which is explained in more detail in the discussion section, where a comparative analysis of the same amounts of data in the GAMINESS environment and the 3DcityDB Postgres/Oracle is made.

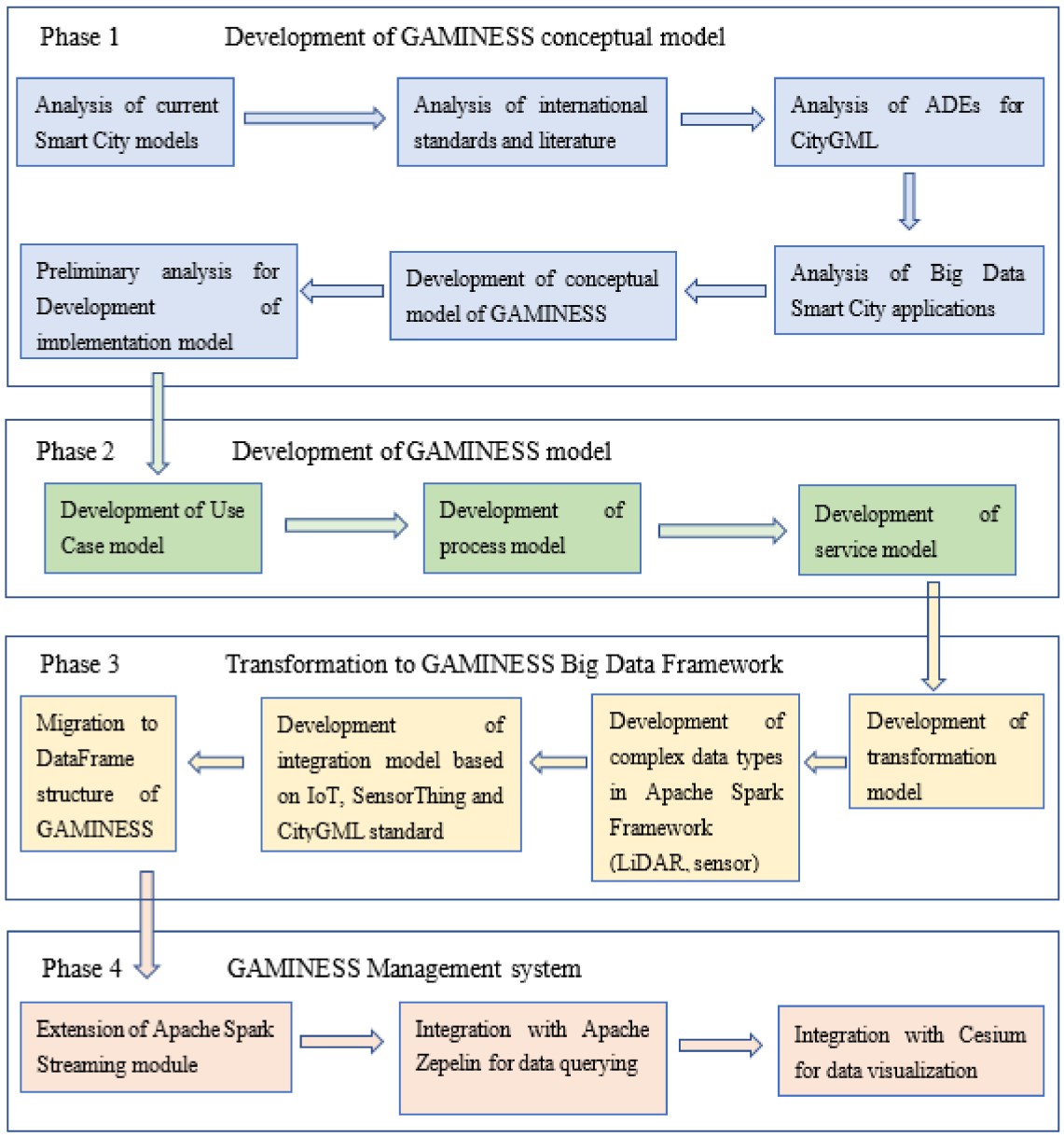

**Figure 2.** A roadmap to adopt the GAMINESS management system.

In smart city-related data processing systems, according to the DigitalTwins model, data appear multiple times with time validity details, meaning that the same data are repeated, leading to more complex data processing. In the GAMINESS management

system, this problem is overcome by using lifespan rules, and only currently valid data are inputs for processing.

The proposed solution provides the possibility of transformation from heterogeneous data sources into the proposed structure, taking into account the relationship between data and data hierarchy in order to control the management of such large amounts of data. This stratification is defined by a data model in which there is a connection between data that are geospatially determined (point cloud combined with sensor observations integrated into a city model). This forms two separate units in the loading model, realized by user type definitions for the classes of point and sensor clouds, which provide loading of complex data types. In this way, the complexity problem is solved. The GAMINESS management system provides better functionality of the system based on the Apache Spark programming language in the MapReduce distributed data processing approach, shown through the five V concept. In the big data environment, the basic concept of MapReduce involves splitting the process into two subprocesses: the reduction phase and the data mapping phase.

There are appropriate algorithms that perform adequate data merging without any loss. One of the most well-known data storage systems in this context is Hadoop, which has no developed geospatial component and is therefore not adequate for use in this research. In the process of data reduction, there are components of HDFS (Hadoop distributed file systems) that break objects into smaller sub-elements, which are located on clusters of network systems. By creating blocks, distribution is performed through the cluster, and in that way, the source data are reduced, and appropriate mapping is performed in the system. The basic abstract phenomenon defined in the Apache Spark environment is the existence of a resilient distributed dataset (RDD), which is a collection of elements that are partitioned through cluster nodes and mapped into this structure, which is unique and provides the possibility of parallelization. In the GAMINESS environment, RDD is created from the Scala collection of input data mapped to the Apache Spark structure. The Scala has full support for functional programming and a very strong system of static types. This allows programs written in the Scala to be very concise and thus smaller in size than other general purpose programming languages [53]. The system enables the persistence of RDD in the system memory, which provides the possibility of reusing the mentioned RDD through parallel operations. The use of such reduced system files results in scalability and shortening of data processing time, which is one of the biggest contributions in structuring smart city data in this model.

With addition of the geospatial component in this management system, smart cities can include different geospatial data that are in different structures, e.g., data collected from geotechnical sensors. In Section 4.3, an example of using the temperature sensors is presented. Measured values are mapped into elements which are part of the geosensor in the distributed environment based on the Apache Spark framework. The system provides reduction and scalability of the three-dimensional space to a one-dimensional array while preserving the spatial location of points with corresponding attributes regardless of their position. This provides the possibility of conection and management of a large number of different sensors with a huge input of data in a small space.

*3.1. Experimental Platform Based on Apache Spark Framework in the Process of Creating the Smart City Management System Model*

Apache Spark is built on the resilient distributed dataset (RDD) abstraction, which distributes datasets in the distributed memory of a cluster. RDDs are resilient to lost tasks or hosts due to their functional programming approach. Apache Spark provides a language-integrated Scala API enabling the expression of programs as data flows of transformations (e.g., map, filter) on RDDs. RDDs are composed of rows of objects and have additional schema information about the data types of each column (19). The collection of RDDs in Apache Spark is defined as DataFrames. RDDs can be explicitly cached by a programmer in memory or on a disk by workers. Fault tolerance is provided by recomputing the sequence of transformations for the missing partition(s). Apache Spark provides four high level

libraries: streaming for micro batch processing, GraphX for graph computation, MLlib for machine learning, which was used by Liu and Boehm [31] for lidar point cloud classification, and Spark SQL. Spark SQL, as Spark's structured data and relational processing module, supports a subset of SQL [54]. Spark SQL as Apache Spark library works with logical and physical relational operators. In this framework, in the context of variety advancement, there is a need to define new types which are going to exceed default as user defined types. User defined types can be used to define the DataFrame schema, and they can be used by Apache Spark to optimize query execution [27,55]. For this reason, in this paper, the Apache Spark framework was selected to develop a model. The model is in accordance with [27].

In order to describe data with all attributes together with smart city elements, a conceptual model of GAMINESS was developed. In the conceptual model, CityGML standard was extended to implement IndoorGML standard and IoT standard. Packages of classes that contained only numerical data had to be extended with user defined types to descibe spatial attributes collected from sensors. Implementation schema of the GAMINESS management system was based on JSON. The transformation module was developed in order to transform the JSON GAMINESS management system to DataFrames on Apache Spark. DataFrames consist of RDD schema where RDDs are composed of rows of objects with additional schema information such as types of data in each column, basic types, and user defined types. GAMINESS provides storage of data in an RDD structure with all their elements, with the possibility to make a new query over stored data. Those SQL queries can use a set of already user-defined functions. The system provides volume and value advancement, which are described throughout the cluster formulation of the processing network. Apache Spark has built-in modules for machine learning components which are intended for complex analysis in the context of the smart city for variability and velocity advantages of the GAMINESS.

### 3.2. Development of GAMINESS Conceptual Model

The conceptual model of the GAMINESS management system was defined as an extension of the existing CityGML model. This standard is described in detail through CityGML specifications [46,56,57]. The conceptual model consists of two submodels: a geospatial and a sensor submodel. The geospatial submodel has an extension for the interior orientation parameters defined by the IndoorGML standard. The sensor submodel is based on the Dinamizers submodel of CityGML standard, which is extended for IoT and SensorThings standards. The CityGML standard is a common information model for 3D urban objects and provides a comprehensive and detailed representation. The standard is defined as XML format for storing objects based on ISO 19,118 with a possibility to share virtual 3D city models. Proposed extensions defined by the GAMINESS management system are in accordance with the OGC (Geography Markup Language—GML3) and the definition of the CityGML 3.0 conceptual model for storage and exchange of 3D city models presented by Kutzner et al. [58]. The geometric and the thematic models are described through the defined structure. For the geospatial submodel which represents imlementation of the 3D city structure, general representation is possible using data input regardless of data collection technology (LiDAR, remote sensing, UAV, GNSS, classic geodetic terrestrial methods, etc.). Considering the difference in the classic storage systems, there is a possibility to import point cloud data divided into more files. Consequently, the system locates files in clusters and provides a parallelization process, and this is the concept of the MapReduce paradigm on big data. Data can be recorded in LAS format, polygon file format (PLY), and delimited text layer format (XYZ). This way, the value objective is provided. Figure 3 shows the component model of the system and the main packages.

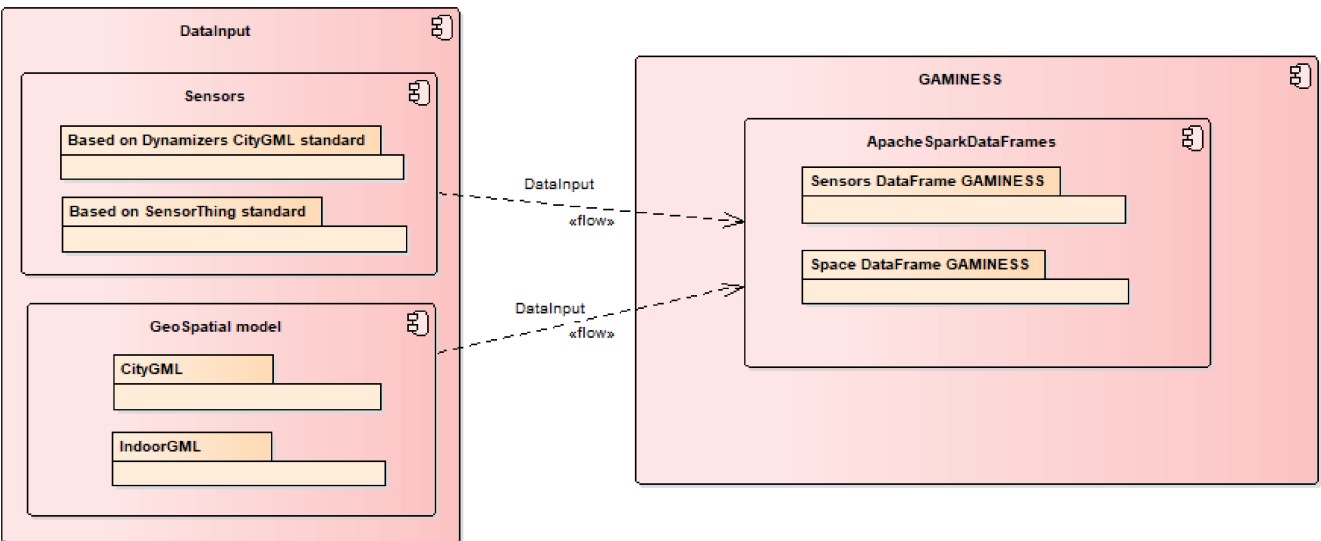

**Figure 3.** System architecture of the GAMINESS components model.

The problem of the variability on input is solved in the geospatial submodel. The point cloud data reader transforms and normalizes input data to the geospatial structure through the calculation of the Morton codes. To make this possible, user defined type was created for reading such data. The sensor submodule was based on the Dynamizers model. The created extensions allow standardized reading of data collected by sensors. The GAMINESS management system provides real-time connectivity between sensors and the system. To solve the variability problem, the basic Dynamizers model is extended with the elements of IoT and SensorML standards. Collected data are sent as JSON format, no matter which type of sensor is on the input (automatized total stations, levels, permanent stations, temperature sensors, etc.). GAMINESS is connected to the OpenADMS platform that has standardized connectivity to different types of sensors and trasfers sensor data as JSON messages. This is how the variety objective is provided.

Van Oosterom et al. [59] benchmarked a number of possible management systems for point clouds and relational data as sensor data in standard relational databases, distributed NoSQL databases, and file-based alternatives. The proposed framework extends the standard MapReduce paradigm for in-memory processing using Hadoop. Therefore, point cloud visualization is commonly known as a point-based rendering. Richter et al. [60] proposed a LiDAR point cloud visualization approach that scales thanks to a pit-of-core approach. Based on these recommendations, the structure of the model is described as a set of data frames which are defined by previously created user defined types and user defined functions. In the next sections, the model is described in regard to conceptual and implementation levels in detail.

*3.3. GAMINESS Logical Data Model with IndoorGML Extension*

CityGML has a common definition of basic entities with attributes and the relationship in a 3D city model, which provides a common feature model of a 3D city space. The basic entity of IndoorGML is a cell, and IndoorGML covers not only properties of each cell but also the topological relationships between cells to explain the structure of indoor space. The geospatial model of CityGML, however, does not sufficiently reflect the properties of indoor space for anything other than a flat interpretation with a single space layer (20). This limitation reflects the basic difference between the aims of two standards: feature modeling and space modeling. Therefore, for the purpose of defining the indoor space as a CellSpace, IndoorFeatures is extended with AbstractFeature (Figure 4). This allows the segmentation of the indoor space layers with possibilities of indoor map selection by area matching, staying, or movement functions on the location. These options provide the opportunity to stay on the GeneralSpace, on the specific cell of the IndoorFeature, and or

to implement movement change according to the cells in the TransitionSpace. This model extension provides the variety objective.

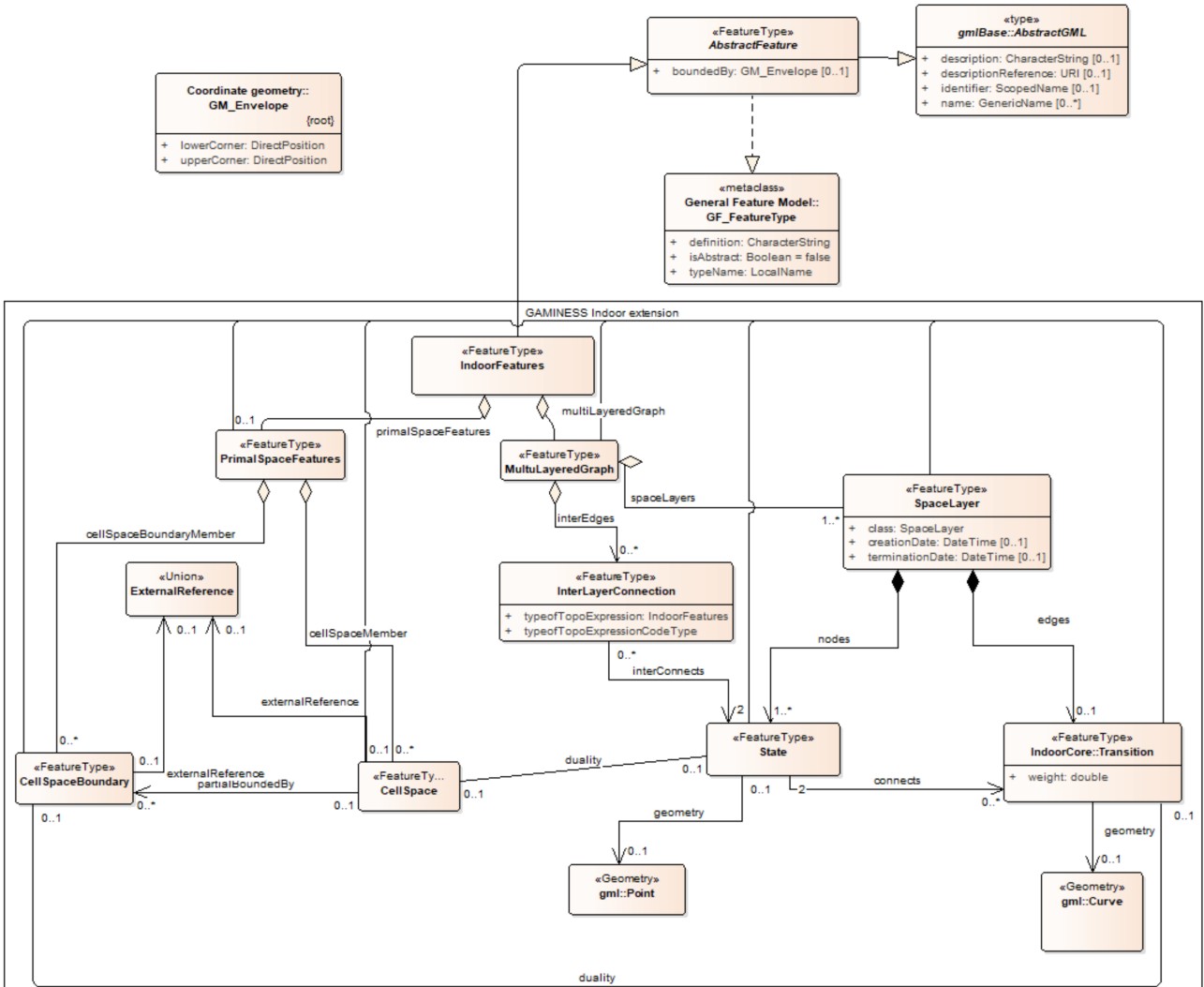

**Figure 4.** GAMINESS basic abstract GML model extended with the IndoorGML navigation classes.

### 3.4. GAMINESS Logica Datal Model with Sensor Extension

The basic CityGML model has initially defined sensors adoption through the Dynamizers package. With the rise of the IoT based collection of sensor data, there is a need for the revision of the model for collecting the observed data. The OGC SensorThings is an open, geospatial-enabled, and unified way to interconnect the Internet of Things devices, data, and applications over the web (Figure 5).

Although it is a new standard, it builds on a rich set of proven and widely-adopted open standards, such as the web protocols and the OGC sensor web enablement (SWE) standards, including the ISO/OGC observation and measurement data model (OGC 10-004r3 and ISO 19156:2011).

In this structure, every device is modeled as a thing entity, which uses location entities to give information about current location and historical location entities for the past trajectory [61]. A thing can have one or more data streams, where each data stream is a logical aggregation of sensor observations, which is produced by the sensor. A data stream group is a collection of observations measuring observation property produced by the sensor mounted on the thing. A data stream has a name, a description of the entity, and a JSON object which contains key-value pairs defined as unit of measurement. The values

of unit of measurement follow the unified code for unit of measure and observation type, used by the service to encode observations. In the proposed model, a thing is the physical sensor with a name, a description, and a JSON object as the properties file. A thing has location, data stream, and historical location.

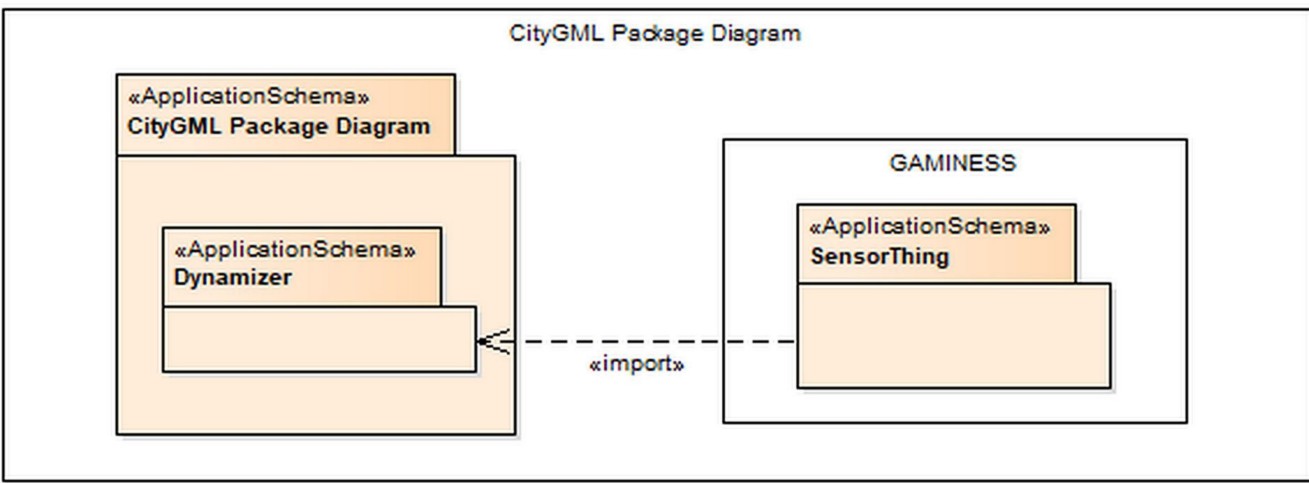

**Figure 5.** GAMINESS CityGML package extension for the SensorThing standard.

The current semantic 3D city models are static in nature and do not support time-dependent properties. The Dynamizer model was extended in order to dynamize features and properties which allow enriching the GAMINESS city model by data from dynamic data feeds (Figure 6). The Dynamizer allows modeling and integrating dynamic properties within the semantic 3D city models. The Dynamizer objects establish explicit links between sensor/observation data and the respective properties of city model objects that are measured by sensors. By making such explicit links with city object properties, the semantics of sensor data become implicitly defined by the city model. The model provides links to observing results via a web feature service (WFS). The things include a WFS link which allows connecting the respective WFS feature with the thing and thus with the related observations. Figure 6 shows the Dynamizers extension with SensorThing standard. This model extension provides the variety objective.

*3.5. Development of GAMINESS Management System*

As mentioned, the GAMINESS management system is a smart city management system with 3D city representation. For the creation of this model, CityGML model is used as an open data model and XML-based format for storage and exchange of virtual 3D city models. The model is based on the application schema for the Geography Markup Language version 3.1.1 (GML3), the extendible international standard for spatial data exchange issued by the Open Geospatial Consortium (OGC) and the ISO TC211 [46]. There are several city management systems that are based on CityGML standard and implemented in different DBMS. 3DcityDB in Oracle and PostgreSQL/PostGIS are subjects of comparison with the GAMINESS management system. In the geospatial submodel, the main improvement is the definition of the interior navigation, using concepts from IndoorGML standard. The sensor submodule is described using the Dynamizers extension for IoT and SensorThings standards. This is used for creating the implementation schema for Apache Spark framework using the big data concept. The model has separate parts as inputs for the geospatial submodel and the sensor submodel, which are later aggregated in GAMINESS DataStore. Figure 7 shows the GAMINESS data-processing module and the whole path of the data from the input to the GAMINESS DataStore.

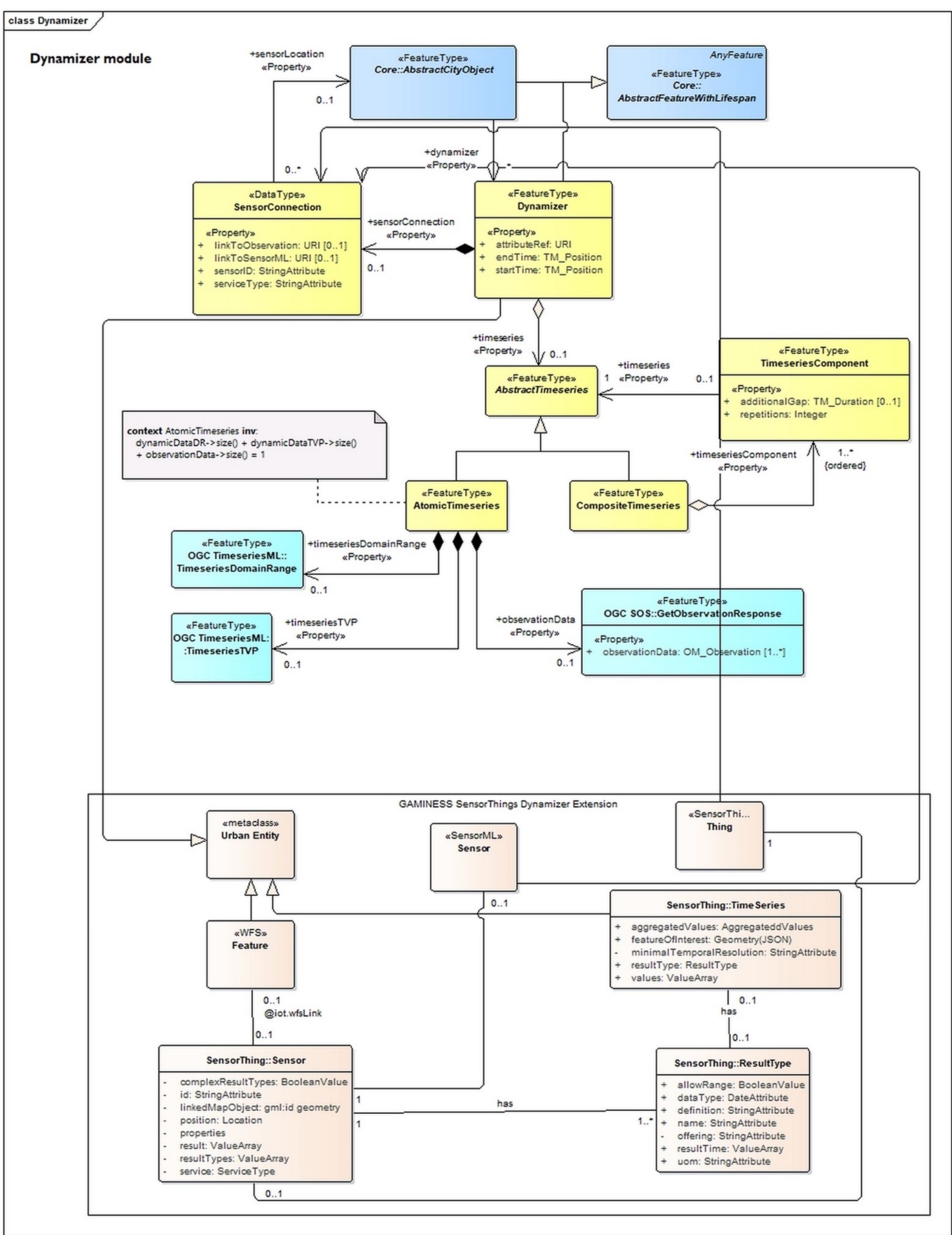

**Figure 6.** GAMINESS Dynamizer with SensorThing optimization.

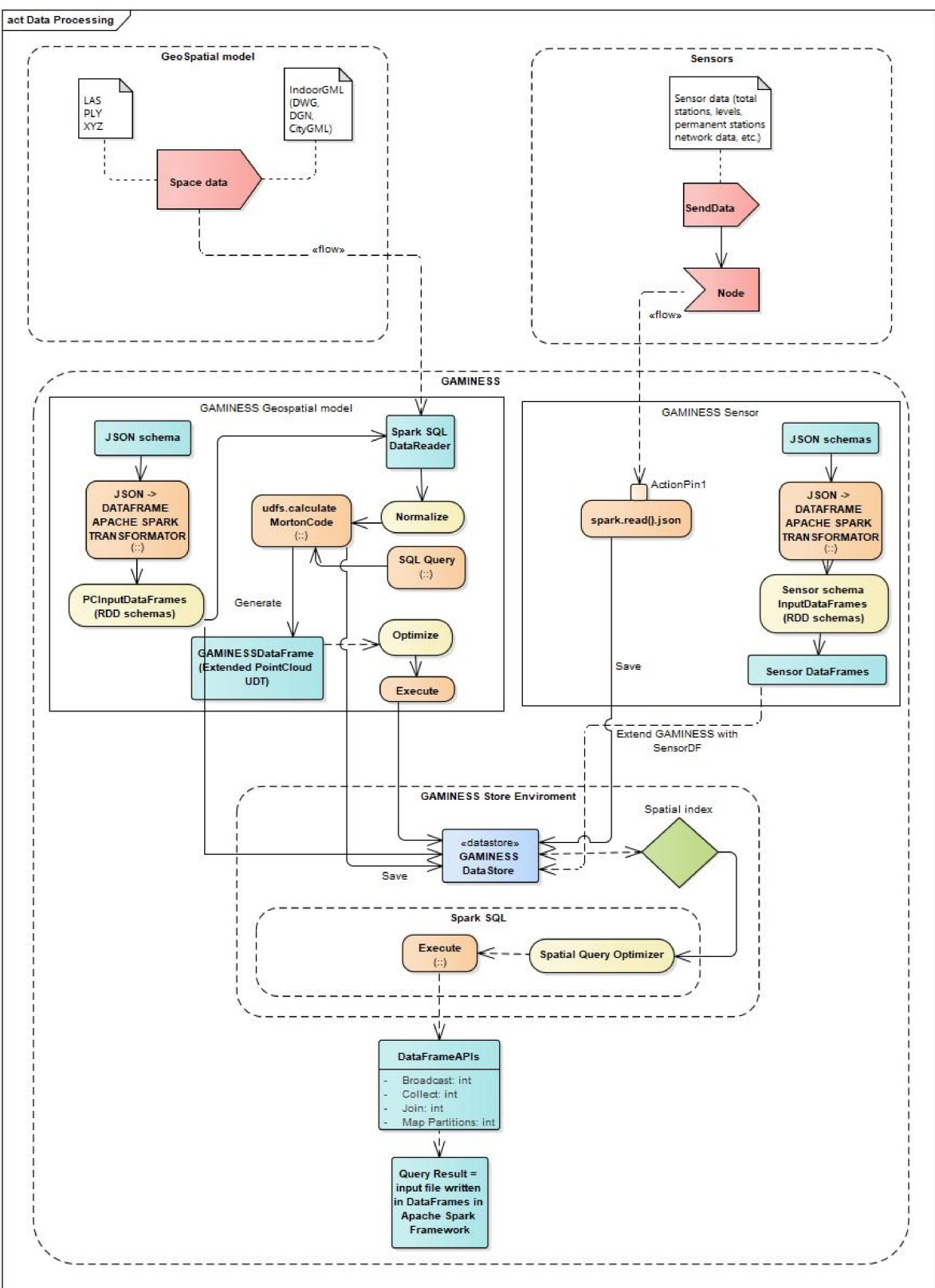

**Figure 7.** GAMINESS data-processing module.

*3.6. Five V in Geosystems*

With the development of technology, there is an increasing use of geospatial data. Almost all the applications used today have a geospatial component in some way. Therefore, geospatial data today are big data and should be accessed as such. There is a whole range

of technical solutions based on different programming languages that are based on big data concepts and provide the possibility of migrating geospatial data to this environment. There is no single implementation solution that works with all data types. Among the most well-known frameworks developed for these needs are:

(a) the Geospark library based on the Apache Spark programming language, which has an extension of the standard RDD to SRDD. It allows one to execute spatial queries using the spatial query processing layer of the GeoSpark class over an object located in the SRDD. It is structured to work with vector data [2].

(b) the Magellan library, which specializes in working with basic geometric primitives and uses the catalyst optimizer to perform spatial merging more efficiently. The library was developed for Python and Scala versions of the Apache Spark programming language. It provides the possibility of performing basic geometric operations such as union, distance, and area, topological analyses through defined operations such as intersects, overlay, etc., and was developed for 2D geometric primitives [62].

(c) Geomesa, which is a distributed space-time library based on the Apache Accumulo column structure; together, they build a library that provides the ability to store geospatial data. It is based on the Scala programming language and provides the ability to perform operations based on the MapReduce paradigm [21].

(d) Geowave, which is a library based on the Apache Accumulo platform and provides multi-dimensional indexing and MapReduce input and output formats for distributed processing and analysis of vector geospatial data [56].

(e) GeoTrellis, which is a library written in the Scala programming language that provides high-speed raster data processing. It is an FOSS cutting edge tool for processing geospatial rasters on the Hadoop platform. With the concept of operation and transformation of input raster data, it belongs to the category of the big data environment [22].

(f) Spark SQL IQmulus library, which provides the ability to read and write point cloud collections in PLY, LAS, and XYZ formats from Spark SQL. The environment is adapted to the Apache Spark version 1.6.2 [63].

The decision to develop an implementation model of the smart city management was made based on the comparative analysis of the existing big data technical solutions. The results of the analysis are shown in Table 1. Since the Magellan geospatial library has best results for the necessary characteristics, it was chosen to be used in the GAMINESS management system.

**Table 1.** Comparative analysis of big data technical solutions for working with geospatial point cloud data.

| Characteristics | Environment | | | | | |
|---|---|---|---|---|---|---|
| | Geospark | Magellan | Geomesa | Geowave | GeoTrellis | Iqumulus |
| Primitive data types—raster | + | − | + | + | + | − |
| Primitve data types—vector | + | + | + | + | + | + |
| Data set types; working with point cloud | − | + | − | − | − | + |
| Support for JSON format | + | + | + | + | + | + |
| Spatial queries | + | + | + | + | + | + |
| Implementation in the Apache Spark programming language | + | + | + | + | + | + |
| Support for the Apache Spark 3.x version | + | + | − | + | + | − |
| Possibility of the implementation on the Apache Spark Scala version | + | + | + | + | + | + |
| Hadoop 2.x | + | + | + | + | + | + |
| Partitioning | + | + | + | + | + | + |
| Coordinate system transformations | − | + | + | | | |
| Geostatistics (spatial interpolation, etc.) | − | − | − | + | + | − |

## 4. Transformation to Gaminess Big Data Framework

As mentioned before, this paper presents the following results:

- analysis between traditional DBMS and big data storage systems in the context of geospatial data;
- extension of the CityGML 3.1.1 standard considering IoT, IndoorGML, and DigitalTwins recommendations;
- development of the conceptual model of a smart city management system which is based on the principles of big data;
- development of a smart city management system model with an integral storage option which does not have a variety problem;
- development of a software package which enables transformation between JSON implementation schema to Apache Spark DataFrame;
- development of new spatial data types based on the Magellan geospatial library (for point cloud, sensors, etc.);
- development of a smart city management system based on big data principles on the Apache Spark platform.

The implementation of the suggested model in the big data framework was performed in the phase where the object-oriented CityGML model was mapped as Apache Spark DataFrames. Apache Spark DataFrame is a big data structure for database schema. In this paper, we used extensions on the Magellan geospatial library for the point cloud data [64]. This extension was developed in previous research described by Pajić et al. [27] and Amović et al. [55], in which a spatio-temporal data model was developed along with an Apache Spark model of point cloud data based on the big data principles. The system provides the import of data (point cloud data and sensor data) based on different standards and structure concepts and their mapping to structures, which is readable by Apache Spark. This way, the system provides variety and volume objectives.

The GAMINESS management system was implemented on the big data concepts in the Apache Spark programming language based on the syntax of the Scala programming language. The basic idea of the solution was to provide an adequate model that provides the possibility of establishing a management system that is defined on the basis of the five V concept. Due to the large amount of data that should provide real-time or near-real-time results, it was necessary to establish a framework based on the MapReduce paradigm that allows distributed processing of geospatial data related to the smart city model, in line with DigitalTwins recommendations. The DigitalTwins concept should provide a copy of the physical world, which allows virtual collaboration, loading data from sensors with fast simulations of conditions for the purpose of accurate result prediction in order to be able to manipulate the physical world. This way, the system provides the variability objective.

The basic need for the DigitalTwins concept is in the virtualization and the optimization of the use of various data that are reflected on the physical world in real- or near real-time. In this way, the possibility of simulating aspects of physical objects and processes is provided [23].

The GAMINESS management system provides the possibility of versioning through appropriate time components, defined in accordance with Inspire recommendations (BeganLifeSpan and EndLifeSpan, with TimeStamp defined type of this data). Based on the previous analysis of available geospatial libraries, which have the ability to work with data in the Apache Spark programming language, elements of the Magellan library were used for the purpose of defining geospatial primitives. Using this environment, a framework for the physical realization of the GAMINESS management system was realized. The Magellan library, which works on the basis of the OGC 19,100 series of standards and defines the possibility of working with basic geometric primitives, was expanded for the purpose of defining the cloud point data type in the original LAS format. This extension was made possible by defining a PointUDT class which, among other things, had a data structure similar to the structured data in LAS format. In this way, all attributes related to the point cloud were retained. MapReduce distributed data storage and processing was performed

by splitting the source files into substructures written in the newly created PointUDT RDD element of the GAMINESS geospatial DataFrame. DataFrame basically forms a database schema over RDD elements. The GAMINESS management system is defined based on two global DataFrames related to the geospatial and the sensor components.

Smart city geosensor networks represent a distributed system that consists of a field of sensors of different types interconnected by a corresponding communication network. The GAMINESS sensor model provides the ability to load measurement data from sensors on the data logger principle which are based on SensorML and IoT standards. Mapping of measurement data converted to JSON messages and streamed to the GAMINESS sensor module was performed. Data were converted to the Apache Spark structure using a complex SensorUDT user defined data type running on the SensorDataRDD model class. The data from the sensor output were brought to the input of the distributed system for estimation. Within the distributed system, based on the available data from the sensors, the most probable information about the phenomenon being monitored was extracted. The GAMINESS sensor module was implemented as data-centric. This means that queries were directed to a region that consisted of a topologically arranged group of cluster sensors. Within a single cluster, there was a single node aggregator, which collected data from the sensor nodes associated with that cluster, analyzed them, aggregated them, and finally transmitted them. One of the advantages of the GAMINESS management system is that, over the established clusters that define separate RDD structures, it is possible to make the appropriate connection after the MapReduce algorithm. The merge provides the possibility of cumulative analysis of local data, which is performed by the node aggregator within the cluster. In this way, the requirements related to communication permeability are reduced. Data aggregation increases the level of accuracy and at the same time avoids data redundancy, which compensates for failures in nodes. The GAMINESS architecture implies that, depending on the decisions, the data can be stored in the file system and used in further analyses. The GAMINESS sensor model provides the possibility of both centralized and decentralized sensor management based on the decisions made in the index part of the GAMINESS management system. This way, the system provides the variability objective. Decisions are returned through the same node through which the data are delivered. In the system, $n$ nodes can be included, and therefore each node is considered as a separate unit. In the system, data are divided into separate RDD elements for each node, thus each answer refers to an individual element. This implies that there is an established system with known node coordinates.

In order for the system to fulfill all the intended functionalities, it was necessary to extend the Apache Spark platform with the basic data types defined in Table 2. These extensions were made as complex user defined types and user defined functions. The following subsection describes how to define UDT and UDF, an experimental platform for the case study with a streaming approach to the model.

**Table 2.** Data type mapping.

| UML | Oracle | PostgreSQL/PostGIS | Apache Spark Framework |
|---|---|---|---|
| String, anyURI | VARCHAR2, CLOB | VARCHAR, TEXT | StringType |
| Integer | NUMBER | NUMERIC | LongType |
| Double, gml:LenghtType | BINARY_DOUBLE | DOUBLE PRECISION | DoubleType |
| Boolean | NUMBER(1,0) | NUMERIC | BooleanType |
| Date | DATE, TIMESTAMP WITH TIME ZONE | DATE, TIMESTAMP WITH TIME ZONE | DateType |
| Primitive Type (Color, TransformationMatrix, CodeType, etc.) | VARCHAR2 | VARCHAR | ShortType |
| Enumeration | VARCHAR2 | VARCHAR | StringType |
| GML Geometry, textureCoordinates | SDO_GEOMETRY | GEOMETRY | PointUDT, MagellanGeometry |
| GML RectifiedGridCoverage | SDO_GEORASTER and SDO_RASTER | RASTER | GEOMESA_RASTER_ BOUNDS_TABLE |
| Texture (only reference of type anyURI in CityGML) | BLOB | BYTE | PointUDT |

### 4.1. User Defined Types and User Defined Functions

The advanced feature in analytical processing in Spark SQL is user defined types. User defined types allow users to create their own classes that are more interoperable with SparkSQL. Creating a UDT for class X provides the ability to create a DataFrame that has a class X in the schema. In order for SparkSQL to recognize UDT, these UDTs must be typed with SQL-defined UDT. For the purpose of creating the GAMINESS transformation model, the Apache Spark framework uses an abstract class, DataType, as the base type of all built-in data types. The standard data types in Apache Spark are described in two main type families, as atomic types and nmumeric types. The atomic types, as internal types, represent types that are not null, user defined types, arrays, structures, and maps. The numeric types are divided on fractional and integral types. In the proposed model, a set of user defined types was created, such as the SensorUDT, to provide functionality of the system:

```
@SQLUserDefinedType(udt=classOf(SensorUDT))
case class Sensor (id: String,
complexResultType: Boolean,
mapObject: IdPointUDT,
time: Timestamp,
position: PointUDT,
properties: String,
result: Array,
service: ServiceType){
class SensorUDT extends UserDefinedType(PointUDT) {
def dataType = StructTypeSeq{
StructField("id", StringType),
StructField("complexResultType", BooleanType),
StructField("mapObject", IdPointUDTType),
StructField("time", TimestampType),
StructField("position", PointUDTType),
StructField("properties", StringType),
StructField("result", ArrayType),
StructField("service", ServiceType)}}

@SQLUserDefinedType(udt=classOf(PointUDT))
case class Point (x:Double,
y: Double,
z: Double,
normx: Integer,
normy: Integer,
normz: Integer,
var mortonCode: Long,
intensity: Integer,
classification: Short,
red: Short,
green: Short,
blue: Short){
class PointUDT extends UserDefinedType(Point) {
def dataType = StructType(Seq(
StructField("x", DoubleType),
StructField("y", DoubleType),
StructField("z", DoubleType),
StructField("normx", IntegerType),
StructField("normy", IntegerType),
StructField("normz", IntegerType),
StructField("mortonCode", LongType),
StructField("intensity", IntegerType),
StructField("classification", ShortType),
StructField("red", ShortType),
StructField("green", ShortType),
StructField("blue", ShortType)}}
```

As a difference to the classical relational DB, the Apache Spark framework allows reading the data directly and writing them in the DataFrame API using Spark SQL library. This enables a columnar SQL handling of Spark's resilient distributed dataset. It also performs SQL execution plan optimizations. The presented code of the UDT class, PointUDT, was elaborated upon in previous research published by Pajic et al. (27). According to Schema, UDT structured type sensor was developed to allow mapping of elements from JSON streaming files to adequate SensorCollectionRDD in the GAMINESS management system. During the data transformation process to the GAMINESS DataFrame sensor submodel, the model uses PointUDT as a type for recognition of streamed point spatial data. After registering this type, a point is recognized as the original object with aggregation of data from Sensor. This object Spark SQL converts to DataFrames, enabling further use of user

defined functions for analytics and predictions. This way, the system provides variety and volume objectives.

### 4.2. Experimental Platforms and Test Enviroment

The case study was performed in the GISLAB at the Faculty of Civil Engineering, Architecture and Geodesy of the University of Banja Luka on four computers connected by a 10 gigabit network. For the comparison in the experiment, 3DCityDB was installed on PostrgeSQL DBMS (version 10.12) and Apache Spark (version 3.0.2).

PostgreSQL was installed on one computer, and Apache Spark was installed and configured on three computers connected as a cluster network, with one master computer (main) and two worker computers. The hardware configuration of computers is given in Table 3.

**Table 3.** Characteristics of used working machines.

| CPU | RAM | HDD |
| --- | --- | --- |
| INTEL Core i7-7700 3.6 GHz | 8 GB, DDR 4, 2400 MHz | 1 TB, 7200 rpm |

### 4.3. Data Desription

Data used in the experiment were classified LiDAR point cloud data and virtualized temperature sensor data provided by the wireless IOT temperature sensor with data logger. The sensor measured temperature in the interval of 1 s and recorded data according to the sensor data definition proposed in the model (Figure 8). Data from the logger using WFS were sent to the GAMINESS.

```
ID,mapObject,Time,Position,properties,result,service
1,sensor1,2020-05-21 15:56:25.843575+00,44.773692 17.210762 196.155456,7F8383EC-D3EC-495C-A8CF-B8BBE85C2920,19.6615830013373,TRUE
2,sensor1,2020-05-21 15:56:26.843575+00,44.773692 17.210762 196.155456,7F8383EC-D3EC-495C-A8CF-B8BBE85C2920,19.6615830013374,TRUE
3,sensor1,2020-05-21 15:56:27.843575+00,44.773692 17.210762 196.155456,7F8383EC-D3EC-495C-A8CF-B8BBE85C2920,19.6615830013373,TRUE
4,sensor1,2020-05-21 15:56:28.843575+00,44.773692 17.210762 196.155456,7F8383EC-D3EC-495C-A8CF-B8BBE85C2920,19.6615830013375,TRUE
5,sensor1,2020-05-21 15:56:29.843575+00,44.773692 17.210762 196.155456,7F8383EC-D3EC-495C-A8CF-B8BBE85C2920,19.6615830013374,TRUE
6,sensor1,2020-05-21 15:56:30.843575+00,44.773692 17.210762 196.155456,7F8383EC-D3EC-495C-A8CF-B8BBE85C2920,19.6615830013374,TRUE
```

**Figure 8.** Visual representation of used data.

### 4.4. GAMINESS Management System (GAMINESS Streaming Aproach)

The 3DCityDB uses the Dynamizers definition for reading sensor data. The basic Dynamizers model has variety of problems reflected through standardisation of the used sensor. In the case study, the temperature sensor adjusted for 3DCityDB was used. GAMINESS provides conectivity to the sensor through the JSON node based on the implementation of two standards mentioned in Section 3.4. Batch processing is not the best possibility for the real-time insights. For the real-time information and processing, we used Apache Spark Streaming for fast, event driven application feedback. The used sensors generated streaming data written as a JSON file, which were processed with Apache Spark and stored in the DataFrames. Sensor DataFrames used Spark Streaming as a direct approach. It worked as a collection of messages where events were organized into categories. They were sorted by producers and sources of data according to schema proposed in the model. Streaming data were partitioned for scalability. Scalability aws achieved by spreading the load across partitions. For the analytics over streamed data, Apache Spark SQL library was used. Spark SQL enables SQL operations on RDDs and external data sources by introducing a DataFrame API which is a columnar database-like view of an RDD. RDD construction is analyzed and optimized at run-time. To enable the reading of a set of streamed files as an Apache Spark Dataframe, a module was created which transformed data from JSON to RDD. Implementation schema of GAMINESS as DataFrames was created by transforming JSON schema elaborated upon in the GAMINESS logical model of data. It was made as a recognition process of JSON fields and mapping to adequate structure fields of RDDs with

data types and UDTs depending on structure type. The conceptual GAMINESS model was exported as a JSON schema. One of the goals of the GAMINESS management system is to support input data integrity when loading JSON data into Apache Spark. For this purpose, there is a possibility of reading an existing JSON-schema file, parsing the JSON-schema and building a Spark DataFrame schema. The generated schema is used when loading JSON data into Apache Spark. This verifies that the input data conforms to the given schema and enables filtering out corrupt input data. This way, data can be grouped to be read in parallel, which is a key for high performance at scale.

The real-time data processing using Spark Streaming performs streaming and batch-processing together using Spark core Api via Spark connector. This way, the system provides volume and value objectives. Basic Spark Streaming divides data streams into more data streams, which are the sequences of RDDs. Each RDD contains the records from a batch interval. An RDD as a distributed collection of elements spreads out across multiple nodes in the cluster. The data contained in RDDs are partitioned, and operations are performed in parallel on the data that were cached.

Any operation applied on a data stream translates to operations on the underlying RDD, which applies transformations to the elements of the RDD. GAMINESS supports fast writing and scaling where tables are automatically partitioned across a cluster by the key range. A table has a write and read function, where read data, written data, and cached column families are available in memory. For the best optimization of the process, the system provides partitioning using Spark Core caching and time minimizing on reads and writes using Spark Streaming.

In the model, it is described how sensor properties files should be configured. This configuration should provide sending messages for partitioning. In this step, GAMINESS provides key-value pair configuration properties for the messages to be sent. A record from the sensor is a key-value pair and is sent to GAMINESS. It consists of topic name, partition number, key, and message value. Type parameters of the record are matched with the serialization properties. For the funcionality of the GAMINESS connected sensor, the initialization of the Spark Streaming context object is made. It creates the DataStream. In the DataStream, each record is written as a line of a text. Message values are parsed into sensor objects, with map operation on the dStream. This map operation applies the Sensor.parseSensor function on the RDDs in the dStream, resulting in RDDs of sensor objects generating sensor data stream RDDs:

```
sensorDStream.fireach RDD { rdd=> val sqlContext =
SQLContext.getOrCreate(rdd.sparkContext)
rdd.toDF().registerTempTable("Sensor")
val res=sqlContext.sql ("SELECT id, mapObject, position, properties, result, service
 FROM Sensor GROUP BY id")
res.show()}
```

The system parses JSON data into sensor Spark classes. The processed RDD registered as a DataFrame is a table which provides the system a possibility to use subsequent SQL statements under the sensor attributes. The sensor RDD objects can be filtered, converted to objects, and then written to the storage system. The receipt of the data must be initiated through the start() function on the Streaming context. Termination stops this process, and then we wait for the streaming computation to finish with the process.

## 5. Discussion

In the paper, we proposed a new GAMINESS management system based on big data principles. The conceptual model of GAMINESS is made using CityGML standard, which is extended for IndoorGML and SensorThing standard elements. The disadvantage of relational databases is that they cannot categorize unstructured data. There is a need to collect and structure those heterogenius data in one model. In this regard, geospatial data are considered along with the new technics for geospatial data aquisition. It is evident that there is a tremendous rise of those data. Additionally, raster data collected using different techniques provide geotagged images with high accuracy and positional resolution. Along

with these are vector data, which are commonly used for the spatial analysis. In the last decade, techniques for massive acquisition of geospatial data are used, where point cloud data are the most common type for representation of geospatial models and complex analysis. All these data require significant computer resources, and there is a need for computers with high performances to accelerate processing vertically, which is reflected on technical and financial aspects of applications, which work with these data. At the level of a smart city, digital and implementation city models can be linked by the Internet of Things, thereby forming an integrated cyber-physical space [32,65].

By definition, smart cities have some physical representation of the space with collection of data that describe different phenomena. Those natural phenomena can be described by the data collected from different sensors, such as geotechincal sensors. For the consideration of the smart city concept, an integration of two submodels was made: indoor navigation and inclusion of sensors without a variety problem in the decision-making process. In the comparasion analysis between the CityGML LoD4 feature model and the IndoorGML cellular space model, the main advantage of the IndoorGML representation is given for the definition of movement pattern and context analysis. The advantages of the CityGML LoD4 model are that it has a higher possibility of visualization and geometric analysis, but the IndoorGML model has better cell finding, hierarchial representation, route definition, route analysis, and context analysis [9]. To host geospatial data features, the Oracle Spatial and Graph platform has a built-in geocoding and routing engine, allowing the quick implementation of customized solutions in the area of routing and navigation. It is worth noting that Oracle Spatial and the built-in routing engine were successfully utilized for routing within an IndoorGML (MLSEM) dataset in related research [66]. The IndoorGML source data could initially be transformed with FME. OpenTripPlanner (OTP) was the OSM routing engine selected for use with our proof of concept implementation. In their research, Jovanović et al. [67] recommended 3DCityDB as a platform for storage of transformed and adopted IndoorGML data, which provides good connectivity to other visualization (GIS and VR) platforms. For the implementation of the geosensors submodule, two standards (IoT and SWE) were considered. SWE is a set of standards which not only allows one to model sensor descriptions and observations but also specifies web services to exchange sensor descriptions and observations in an interoperable way. The combination of open standards (and APIs) eases the delivery of geospatial features and sensor observations in a coherent way and thereby supports interoperable and cross-domain city services [37]. The CityGML model is not able to represent time-dependent and dynamic properties. It allows storing properties as static values. The time-varying properties may be variations of spatial properties, or variation of thematic attributes, or variations with respect to sensor or real-time data. There is a technological gap when it comes to representing changing properties of the city or the city objects [38]. There are many existing open standards defined for unifying interfaces of the SW-IoT service layer. Huang et al. proposed open and interoperable SW-IoT end-to-end architecture based on the OGC SensorThings API. The proposition was bas6ed on three main components of the IoT-PNP: (1) a description file describing device metadata and capabilities, (2) a communication protocol between the gateway layer and the device layer for establishing connections, and (3) an automatic registration procedure for both sensing and tasking capabilities [38].

Stoter et al. concluded that the model of the city is generated independently using different sensor data, reconstruction methods, and software [68]. Resulting models often significantly differ in their geometry, appearance, and semantics. Those models are stored using different formats (XML, graphics, or binary formats), and their underlying data models often also differ. The problem with the CityGML encoded data is that software which supports CityGML is limited. There is a huge number of possible ways in which objects can be defined in CityGML, which makes the full implementation difficult. The 3DCityDB as a database is built upon Oracle Spatial or PostGIS to store the CityGML data model in a relational database.

Another model developed to store city models is CityJSON. This is a format which encodes a subset of the CityGML data model using JavaScript object notation (JSON). The problem with the CityJSON format is the limitation of some objects which are not covered with this model (Table 4). Biljecki et al. [42] said that most of the available 3D city models contain many geometric and topological errors. The most common errors are incomplete surfaces, duplicate vertices, self-intersecting volumes, etc. The GAMINESS management system offers the use of the automatic repair algorithm based on the CityGML topological rules. A relational database uses a structure that allows identification and access to data in relation to another piece of datum in the database. In a relational database, data are organized into tables. In the CityGML model, one or more classes of the UML diagram are often mapped in one table, where the table name corresponds to the class name. Classes are combined into a single table according to the class relations described in the UML class diagram of the model. The Apache Spark framework offers a core for data modeling and processes based on the machine learning algorithms. Apache Spark allows one to leverage the unique ability to simplify machine learning through a single technology. GAMINESS uses Spark Core, which contains the basic functionality of Spark, including components for task scheduling, memory management, fault recovery, interacting with storage systems, and more. Spark SQL provides support for interacting with Spark via SQL. Spark Streaming enables processing of live data stream. MLlib provides multiple types of machine learning algorithms, including binary classification, regression, clustering, and collaborative filtering in addition to supporting functionality such as model evaluation and data import.

**Table 4.** Difference between CityGML and CityJSON data models.

| CityGML Modules and Characteristics Supported | CityGML Modules and Characteristics Not Supported |
| :---: | :---: |
| CityGML Core | |
| Building | |
| Bridge | |
| Tunnel | No support for ADEs |
| CityFurniture | No support for the topological relationships that can be defined, e.g., relative to terrain and relative to water |
| LandUse | No closure surface |
| Relief | No support for arbitrary coordinate reference systems (CRSs). Only an EPSG code can be used. Furthermore, all geometries in a given CityJSON must be using the same CRS. |
| Transportation | In CityGML, most objects can have an ID (usually gml:id), that is, |
| CityObjectGroup | one building can have an ID, but also each 3D primitive forming |
| Vegetation | its geometry can have an ID. In CityJSON, only city object types |
| WaterBody | can have IDs as well as each semantic surface object. |
| Generics | |
| Address Apperarance | |

The 3DCityDB is a relational DB which uses PostgreSQL as DBMS for storing and querying the data. For the purpose of the smart city, there is a need to query large 3D datasets. In the proposed model, import of sensor data in the GAMINESS framework was enabled through the use of specific types defined as user defined types. This is the most important extension in the Spark framework regarding the basic module. The 3D city models are available at different levels of detail and provide adequate storage options for collections of 3D city models. Models should be standardized and structured with their semantic data. Taking into account all analyzed literature and presented discussion, the GAMINESS management system has several advantages:

1.  biG dAta sMart cIty maNagEment SyStem (GAMINESS) is a unique solution that provides the ability to manage a smart city system in respect to big data concepts.

2. Unlike existing solutions (e.g., 3DCityDB), the model expands the representation of the interior space with navigation parameters by integrating the IndoorGML standard into the existing CityGML standard in the Apache Spark environment.

3. The model represents one of the first studies that combines models that take into account the geospatial component which is mapped into the big data platform (the proposed model gives more efficient results of point cloud data processing than the proposed model within the IQUMULUS project).

4. GAMINESS enables both integration of geospatial and data streamed from sensors based on different standards and their translation to the Apache Spark structure with adequate conversion.

5. The model provides all data to be stored in one smart city system, which allows combination and analysis of data on a DBMS level that provides more complex combinations of data. The store on the big data concept provides maximal performance of the existing system and much faster response with decision-making processes. This is not possible in any smart city-based DBMS in the basic relation database.

As a result of this research, a physical model and sensor data are stored together in the GAMINESS management system. GAMINESS provides a better platform for analysis directly from the system storage, thus analysis can be done on the visualisation level, where point cloud data and sensor data are usually connected. The case study explored the effects of joint import of point cloud/sensor data within a cluster framework in regard to the classical import in a relational DB. The import was made on the number of nodes while keeping the data volume fixed. A small dataset was used, consisting of 10 files holding 1,000,000 observations. The model benchmarked Spark implementation of a data import provided by the GAMINESS management system library.

Table 5 presents the comparison between two traditional database management systems (PostgreSQL and 3DcityDB) and the GAMINESS management system regarding the five V problems. In the context of the volume problem, GAMINESS provided better import/process options using big data algorithms and processing over smaller import parts through the system. In the context of the velocity problem, GAMINESS showed better performances. It completed the same task eight times faster than 3DcityDB and four times faster than PostgreSQL. In the context of the variety problem, 3DcityDB and PostgreSQL could import point cloud data, but for other data formats such as sensor data, it would be necessary to adopt new standardization rules. GAMINESS used a user defined types structure which could easily adopt different standardization rules, thus data could be read from data loggers based on different structures. In the context of the variability problem, GAMINESS was extended according to the recommendations of the DigitalTwins model and the Inspire lifespan rules. This ensured that the smart city data had valid time information, while 3DcityDB and PostgreSQL kept only static representations of the physical space of the city. In the context of the value problem, GAMINESS could query data using JSON messages directly on sensors. In 3DcityDB and PostgreSQL, it was possible to query only stored data. The table shows that the GAMINESS management system had better performances in processing data than traditional database management systems. One of the reasons for better results is that GAMINESS allows splitting of the files into clusters and conducts parallel processing of the data.

**Table 5.** Import performances.

| 5 V Parameter | 3DCityDB | PostgreSQL | GAMINESS |
|---|---|---|---|
| Volume | 1 file (21 million) | 1 file (21 million) | 10 files (21 milion) |
| Velocity | 260 ms | 128 ms | 30 ms |
| Variety | Import of the structured point cloud in las format | | Import of the structured point cloud in LAS format/Import data from sensor message and structuring using transformation model to sensor RDD |
| Variability | Static representation of the physical space of the city | | Model defined by the concept of the DigitalTwins (time component defined through lifespan in RDD) |

**Table 5.** *Cont.*

| 5 V Parameter | 3DCityDB PostgreSQL | GAMINESS |
|---|---|---|
| Value | Layer segmentation of input data | Layer segmentation of input data<br>Information extraction from input data which are used to query JSON messages from a sensor |

## 6. Conclusions

In this paper, a new framework for a smart city management system is suggested, which is based on the principles of the big data paradigm. On the conceptual level an improvement of the existing CityGML model was made, which is in compliance with the CityGML 3.3 version and further extensions. Improvements were given for the Dynamizers, where an extension of the existing model for IoT standard was made in the context of a smart city. The GAMINESS management system provides an extension of the basic model of LoD4 for the parameters of IndoorGML standard in the context of interior navigation reflected through better cell finding, hierarchial representation, route definition, route analysis, and context analysis.

Based on the defined conceptual model of the GAMINESS management system, implementation of the system was done in the Apache Spark big data platform. One of the main contributions of this system is a defined model for transformation of the system to the Apache Spark architecture. A library was created with the procedures for transforming the conceptual model, which was exported as JSON schema to adequate DataFrames. To provide that, adequate complex UDTs were developed for reading geospatial data. This provided reading of geospatial data to the GAMINESS framework. The GAMINESS management system is based on the MapReduce alghoritms, which provide better five V parameters regardless of classical RDMBS. In classical DBMS, there is a need to accelerate vertically to provide better performances on the large datasets. On the implementation level, in the GAMINESS management system, continued research was described in Pajic et al., where a model for reading classified point cloud data to Apache Spark framework was developed. This model is used to read and write point cloud data to geospatial DataFrames so they can handle and form physical models of the smart city. Using a similar concept in this research, a method for reading and structuring data from different sensors to a sensor DataFrame was developed. The developed transformation algorithm was used to transform messages from sensors through JSON messages to be read and stored in the sensor RDD. In this system, every sensor is connected on an independant node, thus there is a possibility to make different input conditions on every node through a query which affects the JSON message that is transformed to RDD.

Loading of data scales linearly with the increase in data volume. The native Apache Spark implementation seems to perform best on the maximum data size. The results of the experiment can also be interpreted as data throughput, where millions of points are ingested per second.

The proposed solution can easily be extended with additional operations on point clouds and sensor data through implementation of user defined functions, different segmentations, and feature extractions. The results of the case study show that Apache Spark performs better than the classical relational DB with large sets of point cloud data. The main reason for the better performance is the parallel execution of the query. Due to the horizontal scalability of Apache Spark, the execution times can be improved by adding more computers to the cluster. The future work will include the following:

- new property type creation for different sensor types;
- improvments on nodes module for independent connection of sensors to the system;
- research on using four-dimensional space for dynamic point clouds;
- extending the model in order to enable spatial joins with DataFrames of vector geospatial data;
- development of UDFs with the GUI query builder.

**Author Contributions:** Conceptualization, M.G.; Data curation, M.A.; Formal analysis, M.A., A.R.; Funding acquisition, I.J.; Investigation, M.G.; Methodology, M.A.; Resources, A.R. and I.J.; Validation, M.A.; Visualization, M.A.; Writing—original draft, M.A. All authors have read and agreed to the published version of the manuscript.

**Funding:** This research received no external funding.

**Institutional Review Board Statement:** Not applicable.

**Informed Consent Statement:** Not applicable.

**Data Availability Statement:** Not applicable.

**Conflicts of Interest:** The authors declare no conflict of interest.

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
