# Peer review of "Big Data in Smart City: Management Challenges"

_applsci, doi:10.3390/app11104557_

Round 1
Reviewer 1 Report
The following things make a good impression on me in the newspaper:
- in-depth study of the relevant works on the topic;
- correctly created and shown models and technical differences with the existing ones
- the practical orientation of the development and the demonstrated results, which are supported by cited experimental data.
It would be a good idea to correct some inaccuracies in the paper related to inaccurate or missing cited sources that exist in your list.
Author Response
Dear Reviewer 1,
We would like to thank you for your insightful comments that indeed helped shaping this paper. We have carefully and thoroughly revised the submission, and we believe that we have addressed all comments. To meet the required revision, we have taken all raised issues seriously, which led to modification in the writing.
Here are the responses for your comments.
Point 1: The following things make a good impression on me in the newspaper:
- in-depth study of the relevant works on the topic;
- correctly created and shown models and technical differences with the existing ones
- the practical orientation of the development and the demonstrated results, which are supported by cited experimental data.
It would be a good idea to correct some inaccuracies in the paper related to inaccurate or missing cited sources that exist in your list.
Response 1: We corrected the inaccuracies related to the citing. This is an example, of some citing that is corrected. Other corrections are added in the paper.
Pajic et al described model of a Point Cloud Data Mangement System (PCDMS) fully based on the Big Data paradigm which would allow practically unlimited scalability of the system (Pajic et al, 2018).
Other solution for sensor integration is provided by the 52°North project and there is JavaScript library for visualizing the semantical city model stored in CityGML and the OGC SOS observation service from 52°North (https://52north.org/).

Reviewer 2 Report
I would like to thank the authors for their approach to the Smart Cities management system. I believe that manuscript could of interest to the specific IT specialists.
Despite the interest, there are certain things, which need improvement.
Let’s start with small things:
- The manuscript definitely needs additional spelling (first of all) and English editing.
- Authors must adapt the citing rules of the journal. Seems that the citing in the manuscript and in articles of the journal differs.
- The references with a number of authors should be cited (e.g. Line 60) as Sivarajah et al (one “l” – a bit differently from that is in the manuscript – Sivarajah et all)
- At least the first sentence in the abstract should be rephrased since it is almost identical in https://unu.edu/projects/smart-cities-for-sustainable-development.html
- it would be beneficial to use a uniform definition of the manuscript: paper, paperwork, or work. I would prefer paper.
- Line 153: “This research is structured in sections:” – a doubt that research is structured in section. Most probably research results are presented.
More extensive discussion:
- Lines 375-376: “5V (quantity, speed, variety, variability and complexity)” somehow do not start with the letter “V” and are different from the definition in Line 92
- Lines 475-476: “The paper gives an example of the use of temperature sensors, …”, however, examples were not provided.
- Lines 370-371: “form. Figure 1 describes the advantage of the GAMINESS system …” – honestly, to me, Figure 1 neither showed, nor described the advantages of the GAMENESS system
- The authors have put so many overwhelming details that seem they have lost themselves. I definitely lost myself. The full picture is lost in details. For example, what was the purpose of the present text in Section “2 Related works”? What value-added provides the sentence in Lines 189-191 "The hardware component of computers, including the random access memory (or RAM), central processing unit (or CPU), hard drive, and network controller can be virtualized into a series of virtual machines.”?
- It is not clear what the GAMINESS is about. It is claimed: GAMINESS module; GAMINESS model, GAMINESS conceptual model; GAMINESS Big Data model; GAMINESS system, GAMINESS environment; GAMINESS Management system; GAMINESS city model; GAMINESS implementation model as a system; GAMINESS Big Data framework; new platform called GAMINESS.
- Line 136: “The objectives of this paper are:” the objectives of GAMINESS are presented, but GAMINESS itself is not properly introduced.
- Authors need to work on references. For example, the project “the 52°North project” (Line 318) is mentioned, but where to find more info?
- The results of the validation – how GAMINESS is better than other systems – are put in Table 5 without proper explanation.
In general, authors have put a lot of effort into GAMINESS. The manuscript looks more than an excerpt from the report than a scientific paper. The authors I would recommend clarify for themselves what they would like to tell to the audience and from that start construct the paper.
At the moment it is a chaotic and IT detailed-based manuscript. The GAMINESS as a product/system/… and advantages are not shown.
Author Response
Dear Reviewer 2,
We would like to thank you for your insightful comments that indeed helped shaping this paper. We have carefully and thoroughly revised the submission, and we believe that we have addressed all comments. To meet the required revision, we have taken all raised issues seriously, which led to modification in the writing.
Here are the responses for your comments.
Point 1: The manuscript definitely needs additional spelling (first of all) and English editing.
RESPONSE 1: We did additional spelling and correction of English in the paper. We will send paper for English proofreading in the production phase.
Point 2: Authors must adapt the citing rules of the journal. Seems that the citing in the manuscript and in articles of the journal differs.
The references with a number of authors should be cited (e.g. Line 60) as Sivarajah et al (one “l” – a bit differently from that is in the manuscript – Sivarajah et all)
RESPONSE 2: We corrected the inaccuracies related to the citing and included it in paper.
Point 3: At least the first sentence in the abstract should be rephrased since it is almost identical in https://unu.edu/projects/smart-cities-for-sustainable-development.html
RESPONSE 3: We rephrased the whole abstract. This is rewritten abstract.
Smart City uses digital technologies such as Cloud Computing, Internet of Things or Open Data in order to overcome limitations of traditional representation and exchange of geospatial data. This concept ensures a significant increase in the use of data to establish new services that contribute to better sustainable development and monitoring of all phenomena that occur in urban areas. The use of the modern geoinformation technologies, like sensors for collecting different geospatial and related data, requires adequate storage option for further data analysis. In this paper, we suggest biG dAta sMart cIty maNagEment SyStem (GAMINESS) that is based on the Apache Spark Big Data Framework. The model of GAMINESS management system is based on the principles of the Big Data modelling, which differs greatly from standard databases. This approach provides the ability to store and manage huge amounts of structured, semi-structured and unstructured data in real time. System performance is increasing to a higher level by using the process parallelization explained through the 5V principles of the Big Data paradigm. The existing solutions based on the 5V principles are focused only on the data visualization, not the data itself. Such solutions are often limited by different storage mechanisms and by ability to perform complex analyses on large amounts of data with expected performance. GAMINESS management system overcomes these disadvantages by conversion of Smart City data to Big Data structure without limitations related to data formats or used standard. The suggested model contains two components: Geospatial component and Sensor component that are based on the CityGML and SensorThings standard. The developed model has the ability to exchange data regardless of the used standard or data format into proposed Apache Spark Data Framework schema. The verification of the proposed model is done within the case study for the part of the city of Novi Sad.
Point 4:It would be beneficial to use a uniform definition of the manuscript: paper, paperwork, or work. I would prefer paper.
RESPONSE 4: We corrected this in the paper. We choose the term “paper”.
Point 5: Line 153: “This research is structured in sections:” – a doubt that research is structured in section. Most probably research results are presented.
RESPONSE 5: We corrected the sentence and rewritten the description of paper sections.
“The paper is structured as follows: after the Introduction in Section 1 and related work review in Section 2, Section 3 presents a methodology in a form of roadmap that is used to develop model of GAMINESS management system. Section 4 presents the implementation of GAMINESS into Big Data Framework. In Section 5 the case study for the city of Novi Sad is presented. Conclusions and future work have been discussed afterward.”
More extensive discussion:
Point 6: Lines 375-376: “5V (quantity, speed, variety, variability and complexity)” somehow do not start with the letter “V” and are different from the definition in Line 92
RESPONSE 6: We apologize for this mistake. This sentence in now corrected to contain actual terms that leads to 5V abbreviation.
“5V ( volume, variety, velocity, variability, value)”
Point 7: Lines 475-476: “The paper gives an example of the use of temperature sensors, …”, however, examples were not provided.
RESPONSE 7: We added an explanation in the paper in the subsection 4.3. Wireless IOT Temperature Sensor measurements were used as an input to GAMINESS Geosensor component.
Point 8: Lines 370-371: “form. Figure 1 describes the advantage of the GAMINESS system …” – honestly, to me, Figure 1 neither showed, nor described the advantages of the GAMENESS system
RESPONSE 8: Figure 1 shows how using of this solution could help to solve 5V problems that occurred in other systems, by loading and storing the data from different sources regardless of their structure.
Figure 1 shows possiblities of the proposed GAMINESS management system. It integrates loading and storing data from different sources regardless of their structure according to the 5V rules. Sources are connected to the management system as separate storages. This provides velocity objective. Traditional systems only receive data which are structured according to a predefined concept like CityGML. GAMINESS can recive data from sources without variablity problem. It uses alghoritms to read such structures and map data by using User Defined Types. In traditional systems the speed of transactions directly affect the performance of the system. GAMINESS management system is characterized by the Big Data processing, using cluster network, which speeds up performances vertically and improves overall system performance.
Point 9: The authors have put so many overwhelming details that seem they have lost themselves. I definitely lost myself. The full picture is lost in details. For example, what was the purpose of the present text in Section “2 Related works”? What value-added provides the sentence in Lines 189-191 "The hardware component of computers, including the random access memory (or RAM), central processing unit (or CPU), hard drive, and network controller can be virtualized into a series of virtual machines.”?
RESPONSE 9: We shortened related works for 1.5 pages and it is more concisely defined now. Related works describes 5V problems and solutions find in other author’s researches.
Point 10: It is not clear what the GAMINESS is about. It is claimed: GAMINESS module; GAMINESS model, GAMINESS conceptual model; GAMINESS Big Data model; GAMINESS system, GAMINESS environment; GAMINESS Management system; GAMINESS city model; GAMINESS implementation model as a system; GAMINESS Big Data framework; new platform called GAMINESS.
RESPONSE 10: We harmonized the terminology in the whole paper. GAMINESS is a management system based on Big Data concepts which provide management over different structured, semi – structured and non-structured Smart City data.
We corrected the paper, so when some part of the GAMINESS like its conceptual model, data model, etc. is mentioned, it is clearly emphasized what is it about.
Point 11: Line 136: “The objectives of this paper are:” the objectives of GAMINESS are presented, but GAMINESS itself is not properly introduced.
RESPONSE 11: We clarified in the paper what GAMINESS is about. GAMINESS is a new management system based on Big Data concepts which provide management over different structured, semi – structured and non-structured Smart City data. System consists of two components: Geospatial component and Geosensor component. GAMINESS provides connectivity of different data sources regardless of the standardization rules and data formats. Main advantage of such system is that it represent a solution for the Variety problems in Big Data. GAMINESS also speed up performances vertically, allowing fast responses for the data requested by clients.
Point 12: Authors need to work on references. For example, the project “the 52°North project” (Line 318) is mentioned, but where to find more info?
RESPONSE 12: We added the reference in the paper for the mentioned project. More about this project can be found on https://52north.org/
Point 13: The results of the validation – how GAMINESS is better than other systems – are put in Table 5 without proper explanation.
RESPONSE 13: We added proper explanation of the Table 5. The following paragraph is added to the paper.
Table 5 presents the comparison between two traditional database management systems (PostgreSQL and 3DcityDB) and GAMINESS management system regarding the 5V problems. In the context of the Volume problem, GAMINESS provided better import/process options using Big Data algorithms and processing over smaller import parts through the system. In the context of the Velocity problem, GAMINESS showed better performances. It completed the same task 8 times faster than 3DcityDB and 4 times faster than PostgreSQL. In the context of the Variety problem, 3DcityDB and PostgreSQL can import point cloud data, but for other data formats, like sensor data, it would be necessary to adopt new standardization rules. GAMINESS use UserDefinedTypes structure which can easily adopt different standardization rules so data can be read from data loggers based on different structures. In the context of the Variability problem, GAMINESS is extended according to the recommendations of DigitalTwins model and Inspire lifespan rules. This ensures that the smart city data have validity time information while 3DcityDB and PostgreSQL keep only static representation of the physical space of the city. In the context of the Value problem, GAMINESS can query data using json message directly on sensor. In 3DcityDB and PostgreSQL it is possible to query only stored data. The table showed that GAMINESS management system has better performances in processing data than traditional database management systems. The one of the reasons for better results is that GAMINESS allows splitting of the files into the clusters and conducts parallel processing of the data.
Point 14:In general, authors have put a lot of effort into GAMINESS. The manuscript looks more than an excerpt from the report than a scientific paper. The authors I would recommend clarify for themselves what they would like to tell to the audience and from that start construct the paper.
At the moment it is a chaotic and IT detailed-based manuscript. The GAMINESS as a product/system/… and advantages are not shown.
RESPONSE 14: We corrected the paper to clearly emphasize the advantages of the developed management system. We minimized IT details to make the paper more readable.

Round 2
Reviewer 2 Report
Thank you authors for the updated version of the manuscript.
The comment from the previous review version - "Authors must adapt the citing rules of the journal. Seems that the citing in the manuscript and in articles of the journal differs." - remains valid. Please check https://www.mdpi.com/journal/applsci/instructions on citing.
Lines 77-78: "The goal of this paper is the possibility to handle all „smart data“ in one integral model which connects them and form smart city store with the potential for complex data manipulations". In general, I would doubt, because paper "presents", not "handles".
Line 128: "In the paper, GAMINESS is described." - I just wonder how much information provides this sentence.
Line 136: "The objectives of this paper are:" - I would recommend to find more appropriate wording.
Figure 2, part 4 "Integration wiht Cesium for data visualization"
I would recommend to place Figures in sections and subsections they are indicated, e.g. Figure 6 in subsection 3.4.
Line 625: "this paper deals with the following:" - I would recommend to rephrase this part of the sentence. Paper does not deal. The best - it could "present".
Line 804: "configured on 3 computers, with one master and 2 workers." - This part of the sentence sounds peculiar.
Line 891: "In the paper, the GAMINESS management system is proposed." - I question how much information is presented in the sentence. I would recommend either omit or rephrase.
Line 984: "Using all discusions and related papers," I would recommend to rephrase the part of the sentence, e.g. taking into account all analysed the literature and presented discussion, ...
References should be prepared according to the journal instructions - https://www.mdpi.com/authors/references.
Spelling: please run simple spelling.
Text similarities:
Lines 191-194: "The GeoWave library supports Apache Accumulo and Apache Hbase repositories and provides out-of-the-box support for distributed key-value stores. It uses multiple gridded space filling curves (SFCs) to index data to the desired key-value store." It is taken from https://doi.org/10.3390/ijgi7070265
Lines 203-204 and Lines 206-209: " The methods for classification, feature identification and change detection using large point clouds are described in (Liu et al., 2015, Boehm et al., 2016, Liu et al., " and " Rectangular regions are indexed using the Geohash system and stored in MongoDB database along with the location of a corresponding file. Such a structure allows executions of MapReduce operations on point clouds, either from MongoDB or from an external framework, like Apache Hadoop" are taken from https://doi.org/10.3390/ijgi7070265.
Three full sentences in subsection 3.1 are taken from https://doi.org/10.1145/2882903.2915229
Lines 423-426: " Spark’s structured data and relational processing module, supports a subset of 423 SQL. Spark SQL provides logical and physical relational operators. Spark SQL physical 424 operators use a pipelined iterator model and are implemented as functions applied over 425 the iterator from an upstream operator." It is taken from https://doi.org/10.1145/2882903.2915229
Lines 941-944: " (1) A description file describing device metadata and 941 capabilities, (2) a communication protocol between the gateway layer and the device layer 942 for establishing connections, and (3) an automatic registration procedure for both sensing 943 and tasking capabilities. " It is taken from https://doi.org/10.3390/s19030495
Author Response
Dear Reviewer 2,
Thank you very much for your helpful comments. We have responded to each comment below. The changes also have been highlighted in the paper.
Point 1: The comment from the previous review version - "Authors must adapt the citing rules of the journal. Seems that the citing in the manuscript and in articles of the journal differs." - remains valid. Please check https://www.mdpi.com/journal/applsci/instructions on citing.
Response 1: We corrected the citing and included it in paper.
Point 2: Lines 77-78: "The goal of this paper is the possibility to handle all „smart data“ in one integral model which connects them and form smart city store with the potential for complex data manipulations". In general, I would doubt, because paper "presents", not "handles".
Response 2: We corrected the sentence according to proposition.
The goal of this paper is the possibility to present all „smart data“ in one integral model which connects them and form smart city store with the potential for complex data manipulations.
Point 3: Line 128: "In the paper, GAMINESS is described." - I just wonder how much information provides this sentence.
Response 3: We added an explanation what GAMINESS Management System is. We rewrited this sentence.
In the paper, GAMINESS is described as new management system based on Big Data concepts which provide management over different structured, semi – structured and non-structured Smart City data.
Point 4: Line 136: "The objectives of this paper are:" - I would recommend to find more appropriate wording.
Response 4: We used the term goals instead of the objectives.
The goals of this paper are:
Point 5: Figure 2, part 4 "Integration wiht Cesium for data visualization"
Response 5: We corrected this sentence.
Integration with Cesium for data visualization.
Point 6: I would recommend to place Figures in sections and subsections they are indicated, e.g. Figure 6 in subsection 3.4.
Response 6: We moved Figure 6. to section 3.4.
Point 7: Line 625: "this paper deals with the following:" - I would recommend to rephrase this part of the sentence. Paper does not deal. The best - it could "present".
Response 7: We changed this sentence.
“As mentioned before, this paper presents following results:“
Point 8: Line 804: "configured on 3 computers, with one master and 2 workers." - This part of the sentence sounds peculiar.
Response 8: We changed and additionally explained terms master and worker.
PostgreSQL was installed on one computer and Apache Spark was installed and configured on 3 computers connected as cluster network, with one master computer (main) and 2 worker computers.
Point 9: Line 891: "In the paper, the GAMINESS management system is proposed." - I question how much information is presented in the sentence. I would recommend either omit or rephrase.
Response 9: We changed this sentence.
In the paper we proposed new GAMINESS management system based on big data principles.
Point 10: Line 984: "Using all discusions and related papers," I would recommend to rephrase the part of the sentence, e.g. taking into account all analysed the literature and presented discussion, ...
Response 10: We used proposed part of the sentence.
Taking into account all analysed literature and presented discussion, GAMINESS management system has several advantages:
Point 11: Lines 191-194: "The GeoWave library supports Apache Accumulo and Apache Hbase repositories and provides out-of-the-box support for distributed key-value stores. It uses multiple gridded space filling curves (SFCs) to index data to the desired key-value store." It is taken from https://doi.org/10.3390/ijgi7070265
Response 11: This sentence was taken from our previous research paper (Pajic, V.; Govedarica, M.; Amović, M. Model of Point Cloud Data Management System in Big Data Paradigm. International Journal of Geo-Information. 2018, DOI: 10.3390/ijgi7070265). We added citation.
Point 12: Lines 203-204 and Lines 206-209: " The methods for classification, feature identification and change detection using large point clouds are described in (Liu et al., 2015, Boehm et al., 2016, Liu et al., " and " Rectangular regions are indexed using the Geohash system and stored in MongoDB database along with the location of a corresponding file. Such a structure allows executions of MapReduce operations on point clouds, either from MongoDB or from an external framework, like Apache Hadoop" are taken from https://doi.org/10.3390/ijgi7070265. Model of Point Cloud Data Management System in Big Data Paradigm
Response 12: This sentence was taken from our previous research paper (Pajic, V.; Govedarica, M.; Amović, M. Model of Point Cloud Data Management System in Big Data Paradigm. International Journal of Geo-Information. 2018, DOI: 10.3390/ijgi7070265). We added citation.
Point 13: Three full sentences in subsection 3.1 are taken from https://doi.org/10.1145/2882903.2915229 Big Data Analytics with Datalog Queries on Spark
Response 13: We rewritten parts of this paragraph.
Apache Spark provides 4 high level libraries: Streaming for micro batch processing, GraphX for graph computation, MLlib for machine learning which is used by Liu and Boehm [51] for lidar point cloud classification, and Spark SQL. Spark SQL as a Spark’s structured data and relational processing module, supports a subset of SQL [59]. Spark SQL as Apache Spark library works with logical and physical relational operators. In this framework in the context of Variety advancement, there is a need to define new types which are going to exceed default as User defined types. User defined types can be used to define the DataFrame schema and they can be used by Apache Spark to optimize query execution [34,60]. For this reason, in this paper, the Apache Spark Framework is selected to develop a model. The model is in accordance with [34].
Point 14: Lines 423-426: " Spark’s structured data and relational processing module, supports a subset of 423 SQL. Spark SQL provides logical and physical relational operators. Spark SQL physical 424 operators use a pipelined iterator model and are implemented as functions applied over 425 the iterator from an upstream operator." It is taken from https://doi.org/10.1145/2882903.2915229
Response 14: We added appropriate citation for that sentence (number 59 in the reference list)
Point 15: Lines 941-944: " (1) A description file describing device metadata and 941 capabilities, (2) a communication protocol between the gateway layer and the device layer 942 for establishing connections, and (3) an automatic registration procedure for both sensing 943 and tasking capabilities. " It is taken from https://doi.org/10.3390/s19030495
Response 15: We added appropriate citation for that sentence (number 41 in the reference list)

Round 3
Reviewer 2 Report
Dear Authors,
Thank you for the updated manuscript. In general, I have nothing to add. Just a couple of comments:
- In Line 797 the is an expression "worker computers". Neither Authors nor me (Reviewer) are native English speakers. However, I believe it is a straightforward translation from the mother tongue expression. Would that be something different if it just be computers?
- I would take seriously the warning that some text is taken from other sources even if you are the authors. It might be treated as self-plagiarism. In your case, it is not a severe issue. Certainly, it is the policy of the journal.
I wish the Authors very success in the research.